



# Resolving the roles of soot and dust in cirrus cloud ice formation at regional and global scales: insights from parcel and climate models

Xiaohan Li[1,*], Songmiao Fan[2,*], Huan Guo[2], and Paul Ginoux[2]

[1]Atmospheric and Oceanic Sciences Program, Princeton University, Princeton, NJ, 08540, USA
[2]NOAA Geophysical Fluid Dynamics Laboratory, Princeton, NJ, 08540, USA

**Correspondence:** Xiaohan Li (xiaohanl@princeton.edu) and Songmiao Fan (Songmiao.Fan@noaa.gov)

**Abstract.** Atmospheric aerosols can serve as ice-nucleating particles (INPs), influencing the formation and properties of cirrus clouds. While mineral dust has long been considered an effective INP, the role of soot particles remains less explored, limiting our ability to assess their climate impact. Here we use cloud parcel model simulations to examine the competitive ice nucleation behavior of soot and dust, alongside homogeneous nucleation, under a range of meteorological conditions. These simulations
provide process-level insights into how soot and dust influence cirrus cloud ice formation. To evaluate their large-scale implications, we integrate these results into the GFDL AM4-MG2 global climate model. We find that on the global scale, soot (represented in the model as black carbon, BC) enhance ice crystal number concentration (ICNC) by  5%. However, regional increases are much larger—up to 90% in the upper troposphere (500–250 hPa). The strongest enhancements are observed during boreal spring across Eurasia and the Maritime Continent, and during austral spring over South America and the South
Atlantic. The radiative impacts of BC INPs are also substantial. They enhance the annual global cloud radiative effect in the longwave spectrum by approximately 0.24 W m$^{-2}$ and contribute to statistically significant net warming effect during the polar winter in both hemispheres. These results highlight the distinct roles of soot and dust in cloud ice formation and underscore the need to assess the impacts of rising wildfire emissions on atmospheric ice processes and associated climate effects.

## 1 Introduction

Cirrus are high-level clouds composed primarily of ice crystals, with typical formation altitudes between 8 and 17 km (Lynch et al., 2002). With extensive global coverage of ∼30% in the midlatitudes and up to ∼80% in tropical regions, cirrus clouds have a significant impact on the Earth's climate system and radiation budget (Baran, 2012; Heymsfield et al., 2017). Unlike low-level clouds which typically exert a cooling effect, cirrus clouds are often associated with a net warming of the climate system as they allow most shortwave solar radiation to pass through while efficiently trapping outgoing longwave radiation
(Gasparini and Lohmann, 2016; Storelvmo et al., 2013). However, the magnitude and even the sign of their radiative impact are highly uncertain and poorly constrained (Lynch et al., 2002; Heymsfield et al., 2017). This uncertainty stems largely from the limited understanding of aerosol-cloud interactions, particularly the processes by which aerosols serve as ice nucleating particles (INPs) to alter cloud abundance and radiative properties (Singh et al., 2024; Masson-Delmotte et al., 2021; Li et al., 2024; Lynch et al., 2002; Wang et al., 2014).



Ice nucleating particles are aerosols that facilitate ice formation under conditions where it would otherwise not occur (Hoose and Möhler, 2012; Murray et al., 2012; Li et al., 2024). In the absence of INPs, pure water vapor requires temperatures below -38°C and a relative humidity with respect to ice ($RH_{ice}$) around 150% to form ice crystals through homogeneous nucleation (Koop et al., 2000). The presence of INPs, however, can modify the interfacial water structure and dynamic properties at the particle surface, thereby lowering the energy barrier for the phase transition and enabling ice to form at higher temperatures

or lower ice supersaturation levels through heterogeneous nucleation (Li and Bourg, 2023b, a, 2024). This, in turn, can significantly alter the optical and physical properties of clouds, creating a complex interplay between aerosols, clouds, and climate (Kärcher, 2017; Li et al., 2024; Gasparini and Lohmann, 2016). For example, previous studies have shown that within an optimal range of INP concentrations, cirrus clouds formed predominantly by INP-induced heterogeneous nucleation can contain fewer but larger ice crystals, resulting in shorter lifetimes and a reduced warming effect compared to those formed by homo-

geneous nucleation (Kärcher and Lohmann, 2003; Storelvmo et al., 2013; Mitchell and Finnegan, 2009). However, despite general consensus on the critical role of INPs (e.g. dust) in modulating cirrus cloud properties, identifying the effective INP types and accurately assessing their climate impacts remain controversial and can vary significantly between different studies (Penner et al., 2015; Gasparini and Lohmann, 2016; Storelvmo et al., 2013).

Among various types of potentially important INPs, soot (also known as black carbon, BC) continues to be one of the most

debated components. This debate stems from widely conflicting results reported across multiple scales by laboratory experiments, in-situ measurements, and remote sensing retrievals. For example, laboratory results are generally divergent, with some studies indicating that soot can be an effective INP and its efficiency depends on factors including morphology, particle size, surface oxidation, and prior exposure to water vapor (Kulkarni et al., 2016; Gao et al., 2022; Hoose and Möhler, 2012; Testa et al., 2024). In situ aircraft measurements have previously reported that particles from biomass burning such as black car-

bon are significantly underrepresented in ice residues, leading to the conclusion that effective ice-nucleating elemental carbon particles are of low abundance in the cirrus regime (Cziczo et al., 2013). However, lidar observations have demonstrated that wildfire smoke can trigger cirrus formation (Mamouri et al., 2023). And recent aircraft measurements by NASA's Atmospheric Tomography Missions (ATom) have shown that biomass burning particles in the remote troposphere are dilute but ubiquitous, accounting for approximately one-quarter of the accumulation-mode aerosol number and one-fifth of the aerosol mass (Schill

et al., 2020). Since mineral dust aerosols have been well known as efficient INPs and often coexist with soot (Li and Ginoux, 2025; Deboudt et al., 2010), it is plausible that competition for ice nucleation between dust and soot occurs in mixed aerosol plumes. Mineral dust aerosols are generally more efficient than soot aerosols as INPs, and condensation on ice crystals depletes water vapor. The coexistence of dust and soot and the competitive microphysical processes may mask the signal of soot in certain measurements, contributing to observed inconsistencies. Together, these observations raise an important question:

could soot play a significant role in cirrus cloud formation and influence Earth's radiative budget?

Numerical simulations are promising tools to address this question by providing complementary insights into the role of INPs, connecting observations across different scales, and helping to resolve controversies by linking laboratory processes to cloud-scale dynamics and large-scale climate feedbacks. For example, laboratory measurements are typically conducted under well-controlled conditions of relative humidity, temperature, and pressure (Li et al., 2024). However, in the real atmosphere,



clouds develop through the ascent and expansion of air parcels under variable and often complex meteorological conditions (Heymsfield et al., 2017). To bridge the gap between laboratory studies and in situ cloud observations, cloud parcel models with a Lagrangian framework can be used to simulate the evolution of an individual air parcel as it ascends and descends through the atmosphere. These models explicitly track aerosol activation, competitive ice nucleation among different aerosol species, condensational and depositional growth, and ice crystal sedimentation. By resolving these microphysical processes, parcel

models can predict the meteorological and microphysical evolution of an ice cloud column for a given vertical wind profile, thereby providing a process-level link between laboratory-derived ice nucleating kinetics and real-world cloud formation (Lin et al., 2002; Fan et al., 2017; Kärcher et al., 2022). Similarly, to connect cloud-scale processes to global climate feedbacks, insights from parcel models can be used to inform and constrain cloud microphysics parameterizations within large-scale climate models (Fan et al., 2019). By representing the complex, non-linear interactions between clouds and the broader climate

system, climate models can provide crucial insights for assessing the ultimate impact of INPs and cirrus clouds on the Earth's radiative budget at a global scale. However, a significant gap remains in this modeling framework. To date, there are very few parcel models that explicitly consider competitive ice nucleation between dust and soot, which limits our understanding of soot's behavior in realistic mixed-aerosol environments (Lin et al., 2002; Fan et al., 2017; Kärcher et al., 2022; Yun and Penner, 2012). Moreover, existing climate model studies that include soot effects often rely on oversimplified empirical rep-

resentations, assuming a constant fraction of ice activation at a given temperature and humidity (Zhu and Penner, 2020; Beer et al., 2024; Barahona et al., 2010). Such parameterizations do not adequately capture subgrid-scale cloud processes, where the actual aerosol abundance and competition can strongly influence ice nucleation efficiency. In addition, most existing climate modeling studies primarily focus on the global impacts of soot, without providing a detailed perspective on its regional effects — especially in regions where soot is a dominant aerosol component (Zhu and Penner, 2020; Beer et al., 2024).

To address the aforementioned knowledge gaps, this study presents a systematic framework that combines laboratory data with parcel and climate models to provide insights into the roles of soot and dust as INPs and their resulting climate impacts. First, we use a cloud parcel model driven by laboratory-derived ice nucleation active site density parameterization to simulate the competition between heterogeneous nucleation on dust and soot, as well as homogeneous freezing. Rather than prescribing a fixed ice-active fraction at a given temperature and supersaturation, this setup enables us to predict ice crystal number con-

centrations across a range of aerosol compositions, concentrations, pressure levels, temperature regimes, and updraft velocities representative of cirrus cloud formation. We then incorporate the parcel model results into the GFDL climate model to examine their impacts on the global distribution of ice crystal number concentrations and their interactions with radiation. To assess model performance, these outputs are evaluated against a suite of observational data, including in-situ aircraft measurements and lidar observations. Additionally, we investigate regional effects by focusing on soot- and dust-dominated environments

such as the Tibetan Plateau and the South American outflow region, providing insights into the seasonal and spatial variability of INP impacts. Overall, this integrated approach aims to reduce uncertainties in representing the roles of soot and dust in cirrus cloud formation and their broader climatic implications.



## 2 Method

### 2.1 Parcel model description

#### 2.1.1 Numerical set-up

A comprehensive set of over 5.5 million cloud parcel model simulations was performed to investigate a wide range of conditions relevant to cirrus formation. The parcel model is based on the framework of Fan et al. (2017), but it was adapted to specifically simulate cirrus clouds by applying environmental conditions typical for their formation (De La Torre Castro et al., 2023; Barahona et al., 2017; Fu et al., 2017). The cirrus cloud simulations in this study assume a constant updraft velocity ranging

from 0.1 to 100 cm s$^{-1}$ discretized into 40 bins. The cloud layer is set to a thickness of 300 m; the cloud base pressure is prescribed at values of 100, 200, 300 and 400 hPa; and the cloud base temperature varies between 190 K and 233 K, in 3 K increments. To examine aerosol-cloud interactions, four distinct aerosol types were simulated: soot, dust, sulfate, and sea salt. The mass concentrations for soot and dust spanned from 1 to 10,000 ng m$^{-3}$ across 15 bins, while sulfate and sea salt concentrations ranged from 10 to 1,000 ng m$^{-3}$ using 3 bins each. The parcel model computes pressure (P) and temperature (T)

as an air parcel ascends from its initial state, assuming an adiabatic process. Additionally, the model predicts relative humidity with respect to ice (RH$_{ice}$) and water (RH$_w$), as well as the number concentrations and sizes of droplets and ice crystals. The simulation parameters are summarized in Table A1.

In the parcel model, both homogeneous and heterogeneous nucleation processes are considered (Hoose and Möhler, 2012). Homogeneous nucleation rate is calculated below 238 K following the theory detailed in Koop et al. (2000) for deliquescent

sulfate and sea salt aerosols, as well as liquid droplets formed when the diffusion of water molecules to deliquescent aerosols leads to rapid growth, reaching the critical supersaturation over water (Pruppacher et al., 1998). Heterogeneous nucleation is considered for dust and soot aerosols both above and below 238 K, incorporating both deposition nucleation and condensation-immersion freezing modes. While recent studies suggest that sulfate (Bertozzi et al., 2024), sea salt (DeMott et al., 2016; Wagner et al., 2018), organics (Li et al., 2024; Wolf et al., 2020), and nitrate (Wagner et al., 2020) aerosols may also act as

effective INPs at cirrus conditions promoting heterogeneous ice nucleation, substantial uncertainties remain. Given the scope of this study, which aims to refine the representation of soot and dust ice nucleation, these additional heterogeneous pathways are not explicitly included in the current parcel model. A more comprehensive evaluation of different parameterization schemes incorporating these additional aerosol species would be beneficial in future studies. For the aerosol species considered in this study, lognormal size distribution is assumed. Soot and sulfate aerosols are each represented by a single accumulation

mode, with geometric mass mean diameters of 200 nm and 400 nm, and geometric standard deviations ($\sigma_g$) of 1.7 and 2.0, respectively. Dust and sea salt aerosols are represented by two modes: accumulation and coarse. For dust, 30% of the total mass is assigned to the accumulation mode, while for sea salt, 20% is allocated to this mode. The geometric mass mean diameters for the accumulation and coarse modes are set to 800 nm and 2 $\mu$m, respectively, with $\sigma_g = 2.0$ for both modes.

Once ice crystals form, their subsequent growth is modeled by molecular diffusion and habit evolution, using the two-axis

oblate or prolate spheroid method (Sulia and Harrington, 2011; Pruppacher et al., 1998). The accommodation coefficient for the



condensation of water vapor is set as 0.7, with a value of 0.1 used in sensitivity tests. Ice crystals exceeding a mass-equivalent spherical diameter of 200 µm are excluded from further growth or sublimation, representing their gravitational settling out of the air parcel, as discussed in Fan et al. (2017). Evaporation of liquid droplets and sublimation of ice occur when the air becomes subsaturated. The effects of hydrometeor collision and accretion are not considered in this study. In the parcel model,

as the air parcel ascends to the cloud top, the activated INP numbers are computed using a time step of 1 s. Meanwhile, the ice crystal growth is calculated on an integration time step of 0.02 s, and the pressure, temperature, relative humidity (with respect to both ice and water), as well as the sizes of droplets and ice crystals, are updated every 0.02 s. The integration time step was chosen based on sensitivity analyses to ensure numerical convergence of the model results.

### 2.1.2   Representation of dust and soot INPs

Heterogeneous ice nucleation, including deposition and immersion freezing, occurs on solid aerosol particles at temperatures both above and below 238 K (Hoose and Möhler, 2012). Deposition nucleation occurs below water saturation if water vapor molecules diffuse to aerosol surface and are accommodated (Hoose and Möhler, 2012). Immersion freezing occurs when cloud droplet freezing is catalyzed at the liquid-solid interface (Hoose and Möhler, 2012). It also occurs near water saturation as water condenses in pores and cavities, and subsequently freezes (Marcolli, 2014; Wagner et al., 2016). The treatment of

immersion freezing on mineral dust particles is based on the activity based immersion freezing model (ABIFM) (Alpert and Knopf, 2016). The ABIFM is based on the classical nucleation theory, and formulates the ice nucleation rate as a function of the water activity. The immersion freezing of soot aerosol is neglected in the parcel model (Cziczo et al., 2013). The treatment of deposition nucleation on mineral dust and soot aerosols is based on the parameterizations of Ullrich et al. (2017), which represent the ice nucleation efficiency in terms of the ice nucleation active surface site (INAS). The number of ice crystals

produced by a monodisperse aerosol population is given by

$$N_i = N_{\mathrm{aer}}(1 - \exp(-S_{\mathrm{aer}} \times n_s)) \tag{1}$$

where $N_{\mathrm{aer}}$ is the number density ice nucleating aerosol (cm$^{-3}$), $S_{\mathrm{aer}}$ the aerosol surface area (cm$^2$/particle), and $n_s$ the INAS density (cm$^{-2}$). For a polydisperse aerosol population, the total number of ice crystals is calculated by summing up those calculated for each size bin. The deposition nucleation $n_s$ isolines for desert dust follow U-shaped curves in the ice saturation

ratio–temperature ($S_i$ –T) diagram at temperatures below about 240 K. The negative slope of these isolines toward lower temperatures may be explained by classical nucleation theory, whereas the behavior toward higher temperatures may be caused by a pore condensation and freezing mechanism. The deposition nucleation measured for soot at temperatures below about 240 K also follows U-shaped isolines with a shift toward higher $S_i$ for soot with higher organic carbon content (Ullrich et al., 2017). The deposition nucleation is suppressed by sulfate/soluble coating. The $n_s$ for mineral dust is scaled by a factor of 0.05,

and for soot 0.01, to account for this change of nucleation efficiency (Ullrich et al., 2019).





## 2.2 Climate model description

### 2.2.1 Host model and microphysics scheme

The simulations in this study are performed with AM4-MG2 (Guo et al., 2021), which is based on the Geophysical Fluid Dynamics Laboratory's fourth-generation atmospheric general circulation model, AM4.0 (Zhao et al., 2018a, b). For aerosol representation, AM4-MG2 uses the bulk aerosol scheme embedded in AM4.0, which generates aerosol fields from multiple emission sources, consistent with Zhao et al. (2018a, b). Briefly, AM4.0 simulates the mass distribution of five aerosol species: sulfate, dust, black carbon, organic aerosols, and sea salt. Dust and sea salt are represented with five size bins spanning radii from 0.1 to 10 $\mu$m, while the other aerosol types follow prescribed lognormal distribution in accumulation mode. Aerosol concentrations are calculated based on their emissions (including precursor emissions), chemical production (e.g., sulfate and secondary organic aerosols), transport by advection, and removal processes such as dry and wet (rainout and washout) deposition, as well as convection, as described in detail for AM3 by Donner et al. (2011) and Naik et al. (2013). For cloud microphysics representation, AM4-MG2 replaces the original Rotstayn–Klein (RK) cloud microphysics scheme in AM4.0 with the two-moment Morrison–Gettelman (MG2) scheme, which includes prognostic precipitation (Gettelman and Morrison, 2015a; Gettelman et al., 2015b) as implemented in Guo et al. (2021, 2022, 2025). MG2 prognoses both the mass mixing ratios and number concentrations for four hydrometeor types: cloud water, cloud ice, rain, and snow. To maintain consistency with the prognostic treatment of ice crystal number, AM4-MG2 accounts for the detrainment of ice number concentration from convection to large-scale clouds, following the approach of Kristjansson et al. (2000). The model also considers the shortwave and longwave radiative effects of precipitating hydrometeors (rain and snow).

AM4-MG2 uses the Finite-Volume Cubed-Sphere (FV3) hydrostatic dynamical core (Harris et al., 2020; Lin, 2004), 18 shortwave bands with updated $CH_4$, $N_2O$, and $H_2O$ continuum absorption, and revised formulations for $H_2O$, $CO_2$, and $O_2$ (Paynter and Ramaswamy, 2014). Longwave radiation is calculated using the simplified exchange approximation (Schwarzkopf and Fels, 1991). Convection is parameterized using a "double-plume" scheme, representing coexisting deep plumes (penetrating up to the tropopause) and shallow plumes (generally below 500 hPa), with different lateral mixing rates (Bretherton et al., 2004). The orographic gravity wave drag parameterization accommodates arbitrary topography (Garner, 2018), and the non-orographic component follows Alexander and Dunkerton (1999). Planetary boundary layers are treated using the Lock scheme (Lock et al., 2000), which accounts for down-gradient turbulent diffusion in both convective and stratocumulus regimes. Large-scale cloud fraction is prognosed following Tiedtke (1993). Additional details on AM4-MG2 are provided in Guo et al. (2021, 2022).

### 2.2.2 Implementation of parcel model results

To integrate the process-level insights from the parcel model into the global climate model, we compiled the results from the 5.5 million parcel simulations into a multi-dimensional lookup table. This table parameterizes the number concentration of ice crystals nucleated on dust ($N_{i,\text{dust}}$) and black carbon ($N_{i,\text{BC}}$) as a function of seven input variables: updraft velocity, pressure, temperature, and the mass concentrations of dust, soot, sulfate, and sea salt. Within the GCM at each time step, this lookup



table is queried to determine $N_{i,\text{dust}}$ and $N_{i,\text{BC}}$ when the ambient temperature is below 233.15 K (i.e. -40°C). This threshold was
190 chosen as it aligns with the upper limit of the parcel model's temperature range and focuses the parameterization on conditions
relevant to cirrus formation. A mixed interpolation scheme is used: the GCM interpolates linearly for pressure and temperature,
and logarithmically for updraft velocity and the aerosol mass concentrations.

### 2.2.3 Simulation set-up

Using this framework, we performed Atmospheric Model Intercomparison Project (AMIP) simulations with GFDL AM4-
195 MG2, in which observed sea surface temperature and sea ice were prescribed. AM4-MG2 was run on a cubed-sphere grid with
each face containing 96 x 96 points, corresponding to a nominal horizontal resolution of ∼100 km. The model has 33 vertical
levels extending from the surface to around 1 hPa, with a physical time step of 30 minutes and a dynamic core acoustic time
step of 2.5 minutes.

The simulation was initialized in year 2000 and run through 2006, with the first year treated as model spin-up, and the
200 following 5 years of 2001–2005 for analysis. We note that a five-year period is sufficient to capture stable features of ice crystal
and aerosol climatology over 2001–2005. However, as noted by previous findings (Loeb et al., 2018, 2009), radiation-related
variables such as the cloud radiative effect (CRE) exhibit strong variability and might require longer integrations to reduce
noisy spatial patterns. To examine and ensure the robustness of our radiation analysis, we extended the simulations through
2020 and analyzed the full 20-year dataset for radiation as a comparison with the 2001–2005 analysis. In the main manuscript,
radiation results in Section 3.4 are presented for 2001–2020, and the analysis for 2001-2005 is included in the Supplementary
Information. As will be noted, the two analyses lead to consistent conclusions for radiation, though the extended period provides
more variability and more statistically robust results.

## 3 Results and discussions

### 3.1 Parcel model simulations and process analysis of ICNC

### 3.1.1 ICNC depdendence on meteorological conditions

Figure 1 provides a comprehensive overview of the parcel model simulated ice crystal number concentration (ICNC) as a
function of key meteorological and aerosol parameters. The results presented are for simulations initialized at a constant cloud
base pressure of 300 hPa, with background mass concentrations of 0.1 $\mu$g m$^{-3}$ for both sea salt and sulfate aerosols, which
allows for a systematic evaluation of how ICNC responds to changes in cloud base temperature ($T$), updraft velocity ($w$), and
the mass concentrations of dust ($C_{\text{m,dust}}$) and soot ($C_{\text{m,soot}}$).

The primary meteorological drivers, temperature and updraft velocity, exert strong and systematic control over ice formation.
As shown in Figure 1, the total ice crystal number concentration ($N_{i,tot}$, solid lines) is highly sensitive to $w$, increasing by
several orders of magnitude as $w$ increases from 1 cm s$^{-1}$ (top row) to 50 cm s$^{-1}$ (bottom row). This behavior reflects the fact
that stronger updrafts lead to greater cooling rates, which in turn produce higher peak supersaturations—conditions that activate



a larger number and broader spectrum of INPs. In contrast, the relationship between $N_{i,tot}$ and temperature is non-monotonic. Beginning at the lowest temperatures, $N_{i,\text{tot}}$ initially increases with rising temperature, reaches a maximum, and then declines as temperature continues to rise. This pattern arises from the competition between two opposing processes. Since all simulations are initialized with a relative humidity of 99%, the amount of water vapor available for ice formation is constrained by the low saturation vapor pressure at very cold temperatures. As temperature increases, more water vapor becomes available, supporting

the formation of a larger number of ice crystals. However, at higher temperatures, the thermodynamic favorability for ice nucleation diminishes, reducing the number of activated INPs. This pattern reflects the characteristic U-shaped INAS density curves in the ice saturation ratio-temperature diagram reported by Ullrich et al. (2017). The interaction between increasing water vapor availability and declining nucleation efficiency gives rise to a peak in ICNC at intermediate temperatures ∼200 K. This peak shifts to lower temperatures with increasing updraft velocity, as stronger updrafts make water vapor less of a

limiting factor at colder conditions. Furthermore, the magnitude of the peak ICNC increases substantially with higher INP concentrations (e.g., as $C_{\text{m,dust}}$ increases), reflecting the greater number of available nucleation sites.

These relationships are further detailed by the supplementary figures. The parameter space maps in Figure A1 comprehensively visualize these trends, confirming that the sensitivity to updraft velocity and the non-monotonic dependence on temperature are robust features across the full range of pressures studied. The dependence on updraft velocity $w$ is explicitly

detailed in Figure A2, which shows a near-log-linear increase in ICNC with $w$, although the slope of this log-log relationship slightly decreases at high updraft velocities (e.g., $w > 50$ cm s$^{-1}$). This plateauing effect suggests that as updrafts become very strong, the system transitions from being limited by the availability of water vapor to being limited by the finite number of available INPs. Finally, the modulating effect of ambient pressure is isolated in Figure A3. These plots demonstrate a clear positive relationship between ICNC and pressure $P$ for a given temperature and updraft. While this relationship is near-linear

at lower updrafts, the slope diminishes at higher updrafts (e.g., $w = 50$ cm s$^{-1}$), particularly as pressure increases. This trend is primarily driven by water vapor availability; since simulations begin at a constant relative humidity, parcels at higher ambient pressures contain a significantly greater initial mass of water vapor. However, the plateauing at high updrafts suggests that once the updraft is strong enough to process the abundant water vapor efficiently, the system once again becomes limited by the finite number of available INPs rather than by the water vapor supply.

### 3.1.2 ICNC dependence on aerosol composition

The composition of the aerosol population is a fundamental determinant of the ICNC, with different species playing distinct and competitive roles as summarized in Figure 1 and Figure A1. The contribution from dust ($N_{i,\text{dust}}$) is most prominent at colder temperatures ($T < 210$ K), where it serves as a primary source of ice crystals. This is consistent with the underlying INAS parameterization used in the model, which shows the peak in dust's INAS density occurs around 200 K. As a result, $N_{i,\text{dust}}$

and consequently $N_{i,\text{tot}}$ scale strongly with the initial dust mass concentration ($C_{\text{m,dust}}$) in this colder temperature regime. In contrast, BC acts as a more efficient INP at warmer temperatures ($T > 215$ K), where its INAS density peaks near 220 K. However, a local minimum in $N_{i,\text{BC}}$ is evident near 200 K in Figure 1, which is attributable to strong competition for available water vapor from dust particles at their peak activity temperature. Despite this competition, the sensitivity of BC nucleation to





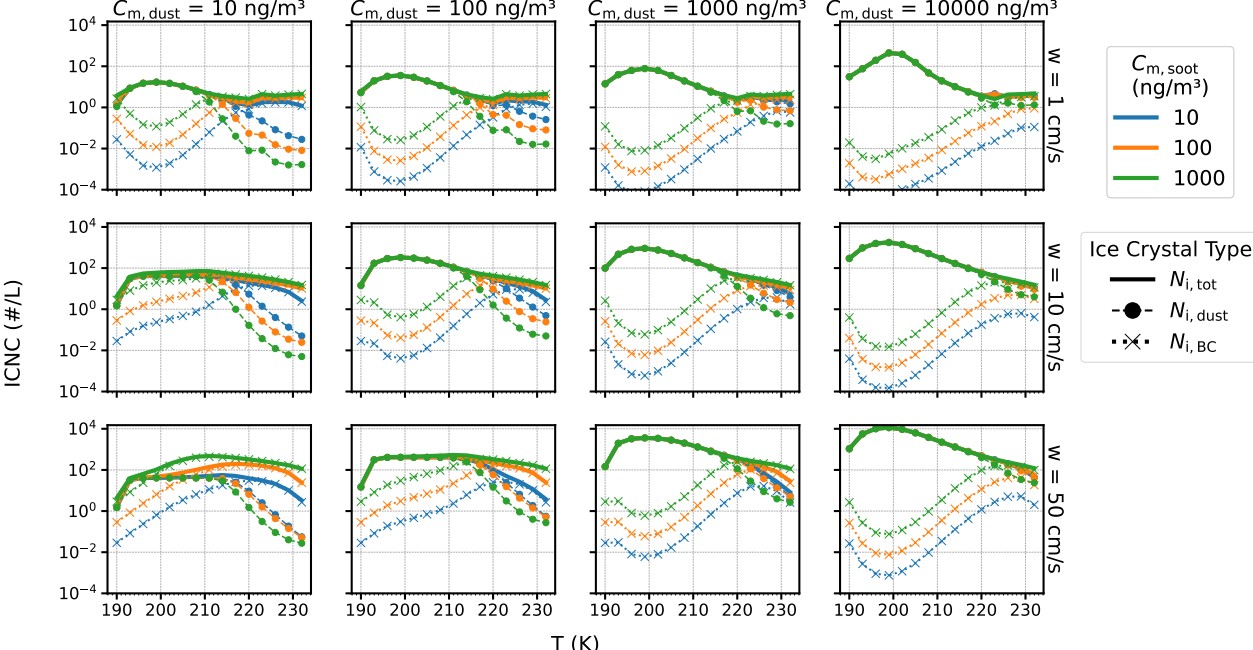

**Figure 1.** Parcel model simulations of ice crystal number concentration (ICNC) as a function of cloud base temperature ($T$). All simulations were initialized with a cloud base pressure of 300 hPa and background mass concentrations of 0.1 $\mu$g m$^{-3}$ for both sea salt and sulfate aerosols. Columns represent varying initial dust mass concentrations ($C_{m,dust}$ = 10, 100, 1000, and 10,000 ng m$^{-3}$), while rows correspond to different updraft velocities ($w$ = 1, 10, and 50 cm s$^{-1}$). Within each panel, line color denotes soot mass concentration ($C_{m,soot}$): blue (10 ng m$^{-3}$), orange (100 ng m$^{-3}$), and green (1000 ng m$^{-3}$). Solid lines show total ice crystal number concentration ($N_{i,tot}$), while circle-dashed and cross-dashed lines indicate contributions from dust ($N_{i,dust}$) and soot ($N_{i,BC}$), respectively.

updraft is notably stronger than that of dust, as indicated by the steeper slope of the $N_{i,BC}$ curve in Figure A2, allowing it to
become a major contributor when high supersaturations are achieved.

To characterize the competition between dust and soot INPs, we analyze the ratio of ice crystals formed on each aerosol type ($N_{i,BC}/N_{i,dust}$). Figure 2 maps this ice number ratio as a function of the initial aerosol mass ratio ($m_{BC}/m_{dust}$) and updraft velocity ($w$) at a cloud base pressure of 300 hPa for a range of temperatures. The results reveal that the competitive balance is highly sensitive to the thermodynamic conditions. Consistent with the temperature-dependent active site densities in the model,
BC is a more effective INP at warmer temperatures ($T > 215$ K), where it can dominate ice formation even at moderate mass ratios (e.g., $m_{BC}/m_{dust} < 1$), particularly at high updrafts. Conversely, at colder temperatures ($T < 210$ K), dust becomes the more prominent INP, requiring a substantially higher BC mass fraction and updraft velocity for BC to contribute equally to the ICNC, as shown by the shift in the $N_{i,BC}/N_{i,dust} = 1$ contour.

The modulating effect of pressure on this competition is detailed in Figure A4, which plots the isolines of the $N_{i,BC}/N_{i,dust}$
ratio for four different ambient pressures. The solid line, representing a ratio of 1.0, marks the critical boundary where the





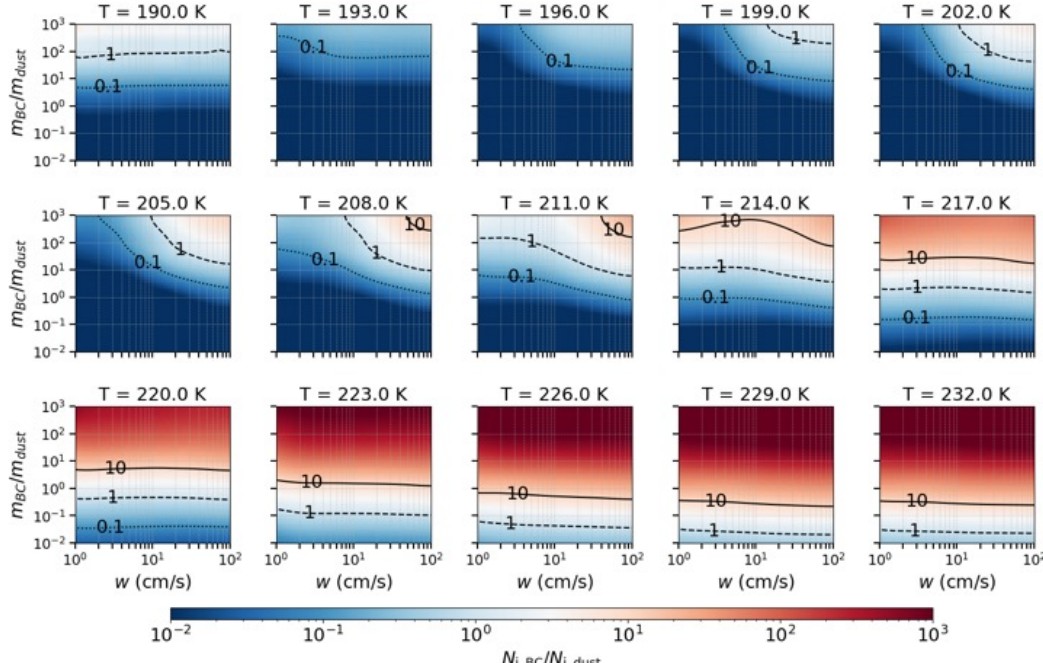

**Figure 2.** The relative importance of BC versus dust INPs, quantified by the ratio $N_{i,\mathrm{BC}}/N_{i,\mathrm{dust}}$ (color scale). The ratio is shown as a function of updraft velocity ($w$) and the initial aerosol mass ratio ($m_{\mathrm{BC}}/m_{\mathrm{dust}}$). Each panel corresponds to a different initial temperature from 190 K to 232 K. Isolines mark where the nucleation ratio is 0.1, 1, and 10. All simulations were conducted with the same cloud base pressure (300 hPa) and fixed background aerosol concentrations of 0.1 $\mu$g m$^{-3}$ for both sea salt and sulfate.

dominant INP type switches. Within any given temperature panel, this boundary shifts to lower $m_{\mathrm{BC}}/m_{\mathrm{dust}}$ and $w$ values as pressure increases from 100 hPa to 400 hPa. This indicates that BC becomes a relatively more effective competitor to dust at higher ambient pressures, a trend that is also visualized in the full parameter space maps shown for 100, 200, and 400 hPa in Figures A7-A9. This pressure dependence again is linked to the greater water vapor mass available at higher pressures for

a fixed relative humidity, which may preferentially benefit the activation of BC over dust. In contrast, the sensitivity of this competitive balance to the background concentrations of sea salt and sulfate is negligible. As shown in Figure A5 and Figure A6, varying these background aerosol concentrations results in almost no change to the activation isolines. In summary, these findings collectively demonstrate that the relative importance of BC and dust as INPs is not fixed but is a complex function of their mass ratio, the updraft velocity, and the ambient temperature and pressure, with little dependence on the background

soluble aerosol concentrations.



## 3.2 Climate model simulations and ICNC climatology

### 3.2.1 Temperature dependence of simulated ICNC

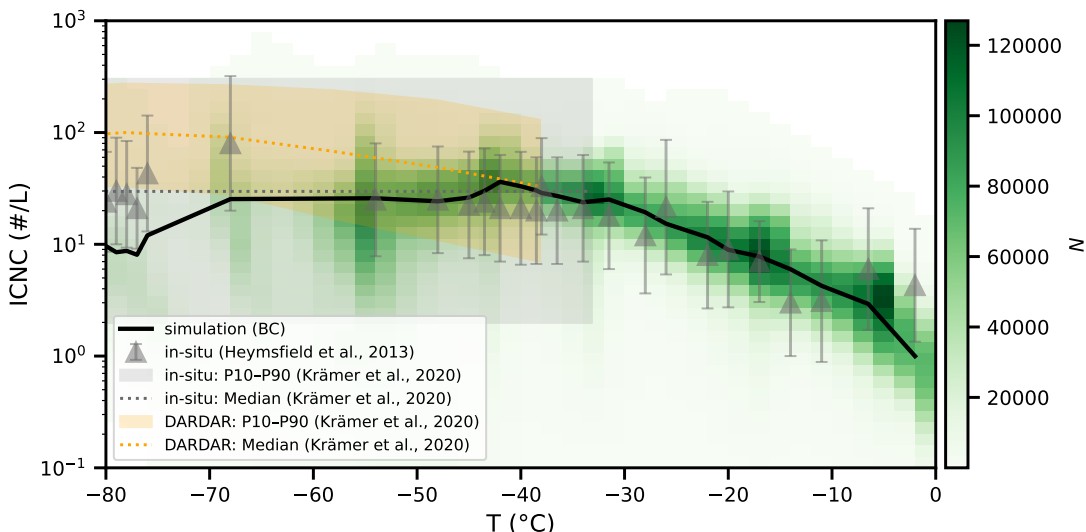

**Figure 3.** Comparison of the parameter space of simulated and observed ice crystal number concentration (ICNC; $L^{-1}$) as a function of temperature (T; °C). The green color scale indicates the data point density ($N$) from the AM4-MG2 simulation with black carbon as ice-nucleating particles, averaged monthly and aggregated globally across pressure levels and time segments for the period 2001–2005. The black solid line shows the median values of these data points in 2 °C temperature bins. For comparison, observational data are shown: gray triangles with error bars show the median and uncertainty of in-situ measurements from various regions summarized by Heymsfield et al. (2013). The gray shaded region and dotted line denote the 10th–90th percentiles and median, respectively, compiled from in-situ observations across multiple regions by Krämer et al. (2020). The orange shaded region and dotted line represent the 10th–90th percentiles derived from 10 years of global DARDAR satellite retrievals tabulated by Krämer et al. (2020).

Figure 3 compares the parameter space of ice crystal number concentration (ICNC) as a function of temperature (T) from AM4-MG2 simulations and observations. The simulation results are monthly mean ICNC values from runs that include black
carbon as ice-nucleating particles, averaged and aggregated globally across pressure levels and time segments for the period 2001–2005. The black line shows the median ICNC in 2 °C temperature bins, while the underlying green shading represents the density of data points within each grid cell. Observational data are drawn from three sources: (1) ICNC measurements compiled by Heymsfield et al. (2013) from multiple in situ campaigns across different regions and time periods, shown as gray triangles with error bars for the mean and uncertainties; (2) the 10th–90th percentiles and median of in situ observations
compiled by Krämer et al. (2020), shown as a gray shaded region with dotted lines; and (3) statistics derived from 10 years of global DARDAR satellite retrievals, also reported by Krämer et al. (2020), shown as an orange shaded region with a dotted line.



The simulated ICNC median spans a range of approximately 1–50 $L^{-1}$ and shows good agreement with in situ observations (gray triangles and shaded percentile range), particularly in the temperature range of $-60°C$ to $0°C$ range. This agreement indicates nice model performance when treating black carbon as ice-nucleating particles. At temperatures below –70 °C, however, the model slightly underestimates ICNC compared to both in-situ medians and DARDAR retrievals. This discrepancy can be attributed to several potential factors, including: (1) contributions from additional INPs neglected in the model; (2) an overly steep negative slope in the U-shaped $n_s$ curves at very low temperatures, where parameters are under-constrained by laboratory observations (Ullrich et al., 2017); and (3) the reduced detectability of thin cirrus clouds at these temperatures in both DARDAR and in situ measurements (Kramer et al., 2020). However, as indicated by the temperature histogram in Figure A10, data coverage in the –80 to –70 °C range is relatively sparse, suggesting that this bias has only limited influence on the overall climatology. Figure A10 also reveals that the difference in ICNC between simulations with and without BC as INPs ($\Delta ICNC = ICNC_{BC} - ICNC_{noBC}$) is on the order of 1 $L^{-1}$ across the cirrus temperature range of $-80°C$ to $-40°C$.

### 3.2.2 Zonal mean distribution and seasonal cycle

Figure 4(a) shows the seasonal and zonal mean distribution of ICNC from the AM4-MG2 simulation with BC treated as INPs. For context, the corresponding seasonal and zonal mean distributions for black carbon and dust are provided in Figure A11. As shown in Figure 4(a), the highest concentrations, exceeding $\sim 10\,L^{-1}$, occur in the upper troposphere. The vertical location of this ICNC maximum exhibits a distinct arch-like pattern with latitude: it occurs at higher pressures (around 500 hPa) in the subpolar regions and rises to lower pressures (approximately 200 hPa) in the tropics. The latitudinal position of this maximum varies seasonally. During the equinox seasons (MAM and SON), the peak concentration is centered near the equator. It shifts into the Southern Hemisphere subtropics during Northern Hemisphere winter (DJF), and moves decisively into the Northern Hemisphere subtropics during boreal summer (JJA).

The impact of including BC as INPs on ICNC is illustrated in Figure 4(b), which shows the absolute difference in ICNC between the simulation with BC and the one without at $T < -40°C$. The primary effect of BC INPs is a notable increase in ICNC, with enhancements reaching up to 10 $L^{-1}$, primarily located in the upper troposphere between approximately 500 hPa and 250 hPa. This region of enhancement is spatially co-located with the ICNC maxima shown in panel (a), indicating that BC most strongly amplifies ice crystal formation in regions where conditions are already favorable for ice nucleation. Figure 4(c) displays the relative enhancement, defined as $\Delta ICNC/ICNC_{noBC}$ at $T < -40°C$. The relative enhancement spatially coincides with the absolute differences, reaching over 50% in many regions and exceeding 90% locally during the MAM season near 40°N and during SON season near 50°S. We note that on the global average, the enhancement is around 5%, suggesting that while the overall effect is moderate, the localized impact of BC acting as INPs can be substantial. We note that some regions in Figure 4(b) show a modest decrease in ICNC, likely because BC INPs depleted water vapor that would have otherwise been transported to form ice elsewhere.

Figure 4(d) provides a global-mean perspective, showing profiles of ICNC and temperature as a function of pressure level to illustrate the overall impact of BC and its relationship with atmospheric conditions. The green solid line represents the globally averaged temperature profile, while the green shaded area marks the pressure levels where temperatures fall below $-40°C$.





**Figure 4. (a)** Seasonal zonal mean of the ice crystal number concentration (ICNC; # L$^{-1}$) from GCM simulations with black carbon (BC) treated as ice-nucleating particles (ICNC$_{BC}$). The x-axis shows latitude, and the y-axis shows pressure (P; hPa). Each panel corresponds to a different season: annual average (All Seasons), December–January–February (DJF), March–April–May (MAM), June–July–August (JJA), and September–October–November (SON). **(b)** Seasonal zonal mean of the difference in ICNC (# L$^{-1}$) between simulations with and without BC as INPs ($\Delta$ICNC = ICNC$_{BC}$ − ICNC$_{noBC}$) at $T < -40°$C. The x- and y-axes are the same as in panel (a). **(c)** Relative change in ICNC compared to simulations without BC as INPs, expressed as the ratio $\Delta$ICNC/ICNC$_{noBC}$ at $T < -40°$C. **(d)** Global-mean profiles of ice crystal number concentration (ICNC) and temperature, plotted using dual y-axes as a function of pressure. The left y-axis shows the globally averaged ICNC (# L$^{-1}$) from GCM simulations with BC as INPs (BC, red solid line) and without BC as INPs (noBC, blue solid line). The right y-axis shows the corresponding global-mean temperature profile (green solid line). The shaded area indicates pressure levels where temperatures fall below $-40\,°$C.





Across all seasons, the globally averaged ICNC from the simulation with BC treated as INPs (red solid line) is consistently higher than that from the simulation without BC (blue solid line) in the upper troposphere, particularly within regions colder than $-40\,°C$. This confirms the widespread enhancement effect of ICNC by BC seen in panel (b). The maximum difference between the BC and noBC simulations occurs near 250 hPa, which also corresponds well with the zonal-mean maxima shown in panel (b). This pressure level is associated with a global-mean temperature of approximately $-40$ to $-60\,°C$, a range that is optimal for cirrus cloud formation.

### 3.2.3 Geographical distribution and regional hotspots

Figure 5 provides a detailed geographical perspective on the distribution of ICNC and the impact of BC at selected pressure levels for different seasons, as shown by the ICNC distribution in panel (a) and the ICNC difference ($\Delta$ICNC) in panel (b). The inset text on each map in panel (a) displays the area-weighted global mean ICNC for simulations with and without BC as INPs, offering a quantitative comparison. Consistent with the previous zonal mean analysis, these maps confirm that the global mean ICNC peaks around the 250 hPa pressure level across all seasons, with average concentrations around $25\,L^{-1}$ and local concentrations reaching up to approximately $100\,L^{-1}$.

The geographical enhancement due to BC, illustrated in panel (b), reveals that its impact is highly regional and strongly influenced by specific meteorological systems. In the Northern Hemisphere, the most pronounced enhancement occurs during the MAM season, across extensive regions of Eurasia and the Maritime Continent. In these regions, the enhancement exceeds $10\,L^{-1}$, and the relative enhancement can locally surpass 50%, consistent with the zonal mean analysis in Figure 4. In the Southern Hemisphere, a distinct pattern emerges during the SON season, where a significant enhancement appears over South America and the adjacent South Atlantic Ocean, extending into the Southern Ocean. Here, the relative increase in ICNC also locally exceeds 50%.

### 3.3 Case studies: analysis of cloud ice formation in selected regions

This section provides a detailed analysis of cloud ice formation in two regions: the Tibetan Plateau and the South American Outflow Region. These regions were selected because they represent the hotspots of BC-induced ice nucleation enhancement in the Northern and Southern Hemispheres, respectively. By focusing on these climatically and geographically distinct regions, we can assess how the impact of BC on cloud ice formation is modulated by different seasonal aerosol regimes and atmospheric conditions.

### 3.3.1 Tibetan Plateau

The first region selected for detailed analysis is the Tibetan Plateau (TP), which spans approximately from 25°N to 45°N latitude and 65°E to 105°E longitude. The TP is a critical region for climate research because it serves as the "Water Tower of Asia", strongly influences regional climate, and is experiencing accelerated warming at nearly twice the global average (Liu and Chen, 2000; Wei et al., 2025). During the Spring (March–April–May, MAM) season, the TP lies within the most prominent





regions of BC-induced ICNC enhancement in the Northern Hemisphere, as identified in Figure 5. These factors underscore the importance of studying its cloud and ice processes, which are explored in detail in Figure 6.

Figure 6(a) presents the spatial distribution of ICNC over the TP during the MAM season at various pressure levels. The top row shows that the highest ICNC values in the BC-included simulation are concentrated over the plateau at mid-to-upper tropospheric levels (e.g., 400 hPa and 250 hPa), with values approaching $100\,\mathrm{L}^{-1}$. The impact of BC is quantified in the subsequent rows. The absolute difference ($\Delta$ICNC, middle row) reveals that the BC-induced enhancement is most prominent over the northern and central parts of the plateau, with increases exceeding $20\,\mathrm{L}^{-1}$ in some areas around 250 hPa. This corresponds

well with the regions of high background ICNC. The relative enhancement ($\Delta$ICNC/ICNC$_{\mathrm{noBC}}$, bottom row) further highlights this impact, showing that BC can increase the local ICNC by over 50% across large areas of the plateau.

The seasonal evolution of these processes, spatially averaged over the TP, is detailed in the time series plots in Figure 6. The simulated mean ICNC (Figure 6(b), blue line) exhibits a broad seasonal high with a bimodal structure: a first peak in spring (MAM) and a second in summer (JJA). Crucially, the ice water path (IWP, Figure 6(d)) follows this same bimodal pattern,

confirming that the changes in crystal number translate directly to changes in total ice mass. This bimodal structure appears to be driven by the sequential influence of the two dominant aerosol types shown in Figure 6(c). The first ICNC and IWP peak aligns with the peak in dust concentration during the pre-monsoon spring, while the second aligns with the peak in BC concentration during the summer monsoon.

Interestingly, the impact of BC on ICNC ($\Delta$ICNC, green line in Figure 6(b)) is strongest during the first ICNC peak in April,

which coincides with the seasonal maximum in IWP, not during the summer when BC concentrations are highest. This indicates that the peak enhancement from BC is not driven by its maximum concentration alone, but rather by the optimal overlap of multiple factors in spring: (1) thermodynamic environment that strongly favors ice nucleation, with rising water vapor and cold atmospheric temperatures; (2) a competitive aerosol regime with a relatively sufficient amount of BC compared to dust, which allows BC to act as an effective INP without being completely out-competed for available water vapor by the more abundant

dust particles. Together, these conditions enable BC to exert a stronger influence on the total ice crystal population in April.

### 3.3.2   The South American outflow region

The second region chosen for analysis is the South American Outflow Region (SAOR), defined here from 35°S to 75°S and 70°W to 45°E. This region is of particular interest as it is strongly influenced by seasonal biomass burning from the South American continent. As shown in Figure 5, the SAOR is the most prominent area of BC-induced ICNC enhancement in the

Southern Hemisphere, particularly during the austral spring (September–October–November, SON). The underlying processes are examined in Figure 7.

Figure 7(a) illustrates the spatial characteristics of ICNC in the SAOR during the SON season. The highest ICNC values in the BC-included simulation are concentrated in a large plume extending from the coast of South America eastward over the South Atlantic Ocean (see also Figure 5(a)), with the highest concentrations found around 250 hPa and 400 hPa. The impact of

BC is most evident in the relative enhancement ($\Delta$ICNC/ICNC$_{\mathrm{noBC}}$, bottom row), which shows a widespread increase of over 50% across the core of the outflow plume at 250 hPa.





The seasonal cycle in the SAOR, spatially averaged, presents a distinct narrative from that of the TP. The simulated mean ICNC (Figure 7(b), blue line) exhibits a bimodal structure, with a first peak in the austral winter (June) and a second, sharper peak in the austral spring (September). This bimodal pattern is also reflected in the IWP (Figure 7(d)). A key feature of this region, in contrast to the TP, is that the LWP is substantially larger than the IWP, indicating a persistent mixed-phase or predominantly liquid cloud environment. These two ICNC peaks appear to be driven by different dominant nucleation mechanisms. The first peak in June occurs when aerosol concentrations are at their annual minimum (Figure 7(c)). This suggests that the high ICNC during this period is likely driven by homogeneous freezing of liquid droplets in a relatively clean, cold environment. The second peak in September, however, aligns perfectly with the dramatic seasonal peak in BC concentration from biomass burning. This indicates a shift to a regime where heterogeneous nucleation on BC particles becomes an important pathway for ice formation. The specific impact of BC (ΔICNC, green line in Figure 7(b)) confirms this interpretation. The enhancement from BC is negligible during the winter but rises sharply to a maximum in September, perfectly in sync with the BC aerosol peak. This demonstrates that while the region supports ice formation year-round, the unique, additional contribution of BC is tightly controlled by the seasonal biomass burning cycle, temporarily making it the primary driver of ice production in the region.

### 3.4  Impact on radiation

As noted in Methodology Section 2.2.3, some previous studies noted that radiation-related variables, such as the cloud radiative effect (CRE), might exhibit noisy spatial patterns and require longer integrations to reduce them (Loeb et al., 2018, 2009). Hence, for cloud radiative effect analysis in this section, we extended our simulation from 2005 to 2020 and analyzed both the radiation results from 2000–2005 and 2000–2020 as a comparison. This analysis was performed to examine both the sensitivity of the radiation statistics to the length of the analysis period and the robustness of our conclusions regarding statistical significance. In this section, the radiation results presented will be from the 20-year simulation (2000–2020), unless otherwise noted. The 5-year default analysis is shown in Supplementary Information Figure A12 and will be discussed in brackets when presenting the 20-year results. As will be shown, the 5-year and 20-year analyses yield consistent results, although the 20-year data show greater variability.

Figure 8(a) shows the geographical distribution of simulated seasonal and annual mean total cloud radiative effect (CRE), separated into its shortwave (CRE$_{SW}$) and longwave (CRE$_{LW}$) components at the top of the atmosphere (TOA) over 2001-2020 (see Figure A12(a) for 2001-2005). The CRE is calculated as the difference between clear-sky and all-sky radiative fluxes at the top of the atmosphere. Specifically, CRE$_{SW}$ is defined as the difference between clear-sky and all-sky upwelling shortwave radiation ($swup_{toa,clr} - swup$), where negative values indicate a cooling effect from clouds reflecting incoming solar radiation. CRE$_{LW}$ is defined as the difference in outgoing longwave radiation ($olr_{clr} - olr$), where positive values represent a warming effect due to clouds trapping outgoing longwave radiation. As shown in Figure 8(a), CRE$_{SW}$ is negative globally, indicating a cooling effect from cloud-reflected solar radiation. This effect is most pronounced over the midlatitude storm tracks and tropical convective regions, with an annual global mean of -48.49 W/m$^2$ (5-year: -48.61 W/m$^2$). Strong seasonal variability is observed, with the greatest cooling occurring during DJF season, reaching -52.88 W/m$^2$ (5-year: -52.92 W/m$^2$). In contrast, CRE$_{LW}$ is





consistently positive, signifying a warming effect. This warming is strongest over regions with extensive high-altitude cloud cover, such as the tropical warm pool. The annual global mean $CRE_{LW}$ is 22.21 W/m$^2$ (5-year: 22.29 W/m$^2$), with minimal seasonal variation. We note that compared with observations, the model exhibits systematic biases, producing a weaker global mean LW CRE and a stronger SW CRE as documented in Guo et al. (2025). Despite these biases, the 5-year and 20-year simulations show consistent results, differing only slightly.

Figure 8(b) illustrates the impact of BC as INPs on CRE, quantified as $\Delta CRE = CRE_{BC} - CRE_{noBC}$ over 2001-2020 (see Figure A12(b) for 2001-2005). A negative $\Delta CRE_{SW}$ means that BC INPs enhance the cloud cooling effect from solar reflection, while a positive $\Delta CRE_{LW}$ indicates an enhanced warming effect from trapping longwave radiation. The spatial patterns of maximum longwave warming ($\Delta CRE_{LW}$) and maximum shortwave cooling ($\Delta CRE_{SW}$) are highly correlated. These regions of strong radiative response directly correspond with the areas showing the largest change in ICNC, as presented in Figure 5. This collocation provides a consistent signal linking the impact of BC on cloud microphysics to the subsequent changes in radiative properties. The strongest effects occur at the previously identified ICNC hotspots, where BC-induced cloud radiative cooling can exceed -8 W m$^{-2}$ (5-year: -10 W m$^{-2}$), and localized warming can reach +8 W m$^{-2}$ (5-year: +10 W m$^{-2}$).

A statistical analysis of the regional and seasonal $\Delta CRE$ is shown in Figure 8(c) (see Figure A12(c) for 2001-2005). The globe is divided into seven latitude bands: Northern High Latitudes (60°–90°N), Northern Midlatitudes (35°–60°N), Northern Subtropics (23.5°–35°N), Tropics (23.5°S–23.5°N), Southern Subtropics (23.5°–35°S), Southern Midlatitudes (35°–60°S), and Southern High Latitudes (60°–90°S). For each band, the radiation data are first spatially averaged by month and then aggregated by season. Boxplots illustrate the distribution of the seasonally and spatially averaged annual $\Delta CRE$ in the shortwave (blue), longwave (orange), and net (green) components. Filled boxplots indicate that the mean $\Delta CRE$ is statistically significant different from zero (p < 0.05). As indicated in Figure 8(c), although the cloud radiative effect induced by BC INPs ($\Delta CRE$) is subject to the well-documented uncertainties of aerosol-cloud interactions, our simulations reveal several statistically significant patterns. Globally, the longwave component ($\Delta CRE_{LW}$) shows a consistent warming signal that is statistically significant across all seasons. This longwave component results in an annual global mean warming of $0.24 \pm 0.06$ W/m$^2$ (5-year: $0.23 \pm 0.04$ W/m$^2$), with seasonal means of 0.19, 0.31, 0.27, and 0.18 W/m$^2$ (5-year: 0.21, 0.30, 0.23, 0.18 W/m$^2$) for DJF, MAM, JJA, and SON, respectively. On a global scale, the short- and long-wave radiative effects due to BC-nucleated ice crystals nearly cancel over an annual cycle. In contrast, distinct regional patterns emerge for the net effect, particularly in the high latitudes. A key finding is the statistically significant net warming effect induced by BC INPs ($\Delta CRE_{net} > 0$, $p < 0.05$) that occurs during the polar winter of each hemisphere (consistent with the 5-year analysis). In the Northern High Latitudes (N. High-latitudes) during the DJF season, the positive $\Delta CRE_{LW}$ (warming) outweighs the negative $\Delta CRE_{SW}$ (cooling) (consistent with the 5-year analysis). Similarly, in the Southern High Latitudes (S. High-latitudes) during the JJA season, a statistically significant net warming is also observed (consistent with the 5-year analysis). These findings highlight the importance of BC as INPs in modulating the polar climate, particularly during the coldest and darkest seasons. Detailed spatial patterns of the net CRE difference induced by BC INPs ($\Delta CRE_{net}$) can be found in Figure A14.



## 4 Conclusions

The role of soot as INPs in cirrus cloud formation has been a long-standing source of uncertainty, complicating efforts to accurately represent aerosol-cloud interactions in climate models. This study employed a dual-scale modeling approach, combining detailed process-level simulations from a cloud parcel model with global simulations from the AM4-MG2 climate model, to systematically investigate the competitive ice nucleation between dust and BC and quantify its large-scale impacts.

Our parcel model simulations, spanning over 5 million unique scenarios, revealed that the competition between dust and
BC is a complex function of the full thermodynamic state and aerosol loading. We found that BC is a more effective INP at warmer cirrus temperatures ($T > 215$ K), while dust dominates at colder temperatures ($T < 210$ K), a behavior consistent with the temperature-dependent active site densities of each species. The relative importance of each INP type is determined by a sensitive interplay between their mass ratio, the updraft velocity, and the ambient pressure, which modulates water vapor availability and thus the competitive balance.

When these process-level insights were incorporated into the AM4-MG2 climate model, the resulting simulations of ICNC showed strong agreement with in-situ and satellite-derived climatologies. While treating BC as INPs resulted in a modest global annual mean ICNC increase of approximately 5%, its impact was highly concentrated in specific regions and seasons, with local enhancements exceeding 90%. Significant "hotspots" of BC-induced ICNC enhancement were identified over extensive regions of Eurasia and the Maritime Continent during the Northern Hemisphere spring (MAM), and over South America and
the South Atlantic during the Southern Hemisphere spring (SON). Our analysis of these regions, particularly the Tibetan Plateau and the South American biomass burning outflow, confirmed that these enhancements are driven by the seasonal overlap of high BC concentrations with favorable meteorological conditions.

The climatic consequences of these microphysical changes are significant. The inclusion of BC as INPs produces a statistically significant global annual mean longwave cloud radiative warming of $+0.24\pm0.06$ W m$^{-2}$. This warming is not uniformly
distributed but is most pronounced in the regional ICNC enhancement hotspots. Notably, our results show a statistically significant net warming in the high latitudes during their respective polar winters.

In summary, this work demonstrates that while dust remains a critical INP, soot from sources such as biomass burning and fossil fuel combustion plays a significant and geographically distinct role in cirrus formation and regional climate than is often assumed. The findings underscore the necessity of moving beyond simplified parameterizations and incorporating detailed,
process-based representations of aerosol competition into climate models. Accurately capturing the effects of BC as an INP is crucial for understanding regional climate dynamics and for projecting the future climate impacts of rising wildfire emissions and other anthropogenic aerosol sources. Finally, we note that recent studies suggest that sulfate (Bertozzi et al., 2024), sea salt (DeMott et al., 2016; Wagner et al., 2018), organics (Li et al., 2024; Wolf et al., 2020), and nitrate (Wagner et al., 2020) aerosols may also act as effective INPs at cirrus conditions promoting heterogeneous ice nucleation. A more comprehensive
evaluation of different parameterization schemes incorporating these additional aerosol species would be beneficial in future studies.





**Figure 5. (a)** Seasonal global maps of ICNC at various pressure levels, from AM4-MG2 simulations where BC is treated as INPs. Text inset on each map provides the area-weighted global mean ICNC ($L^{-1}$) for simulations with and without BC as INPs at the certain pressure level. **(b)** Difference in ICNC ($L^{-1}$) between simulations with and without BC as INPs ($\Delta$ICNC = ICNC$_{\text{BC}}$ − ICNC$_{\text{noBC}}$) at different pressure levels and seasons.



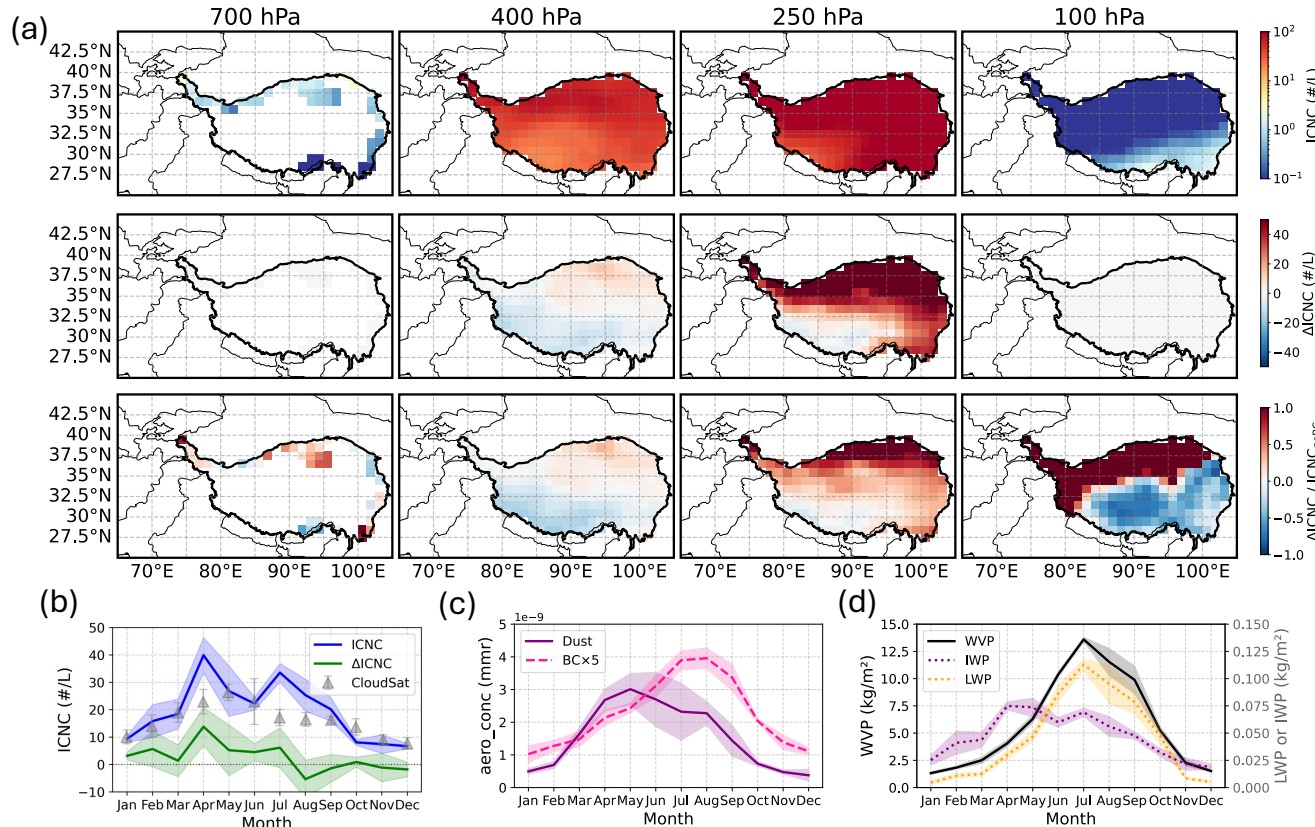

**Figure 6. (a)** Spatial distribution of ICNC during the spring (MAM) season over the Tibetan Plateau (TP). The first row shows the ICNC from the simulation including BC as INPs at various pressure levels. The second row displays the absolute difference in ICNC ($\Delta$ICNC) between the simulations with and without BC as INPs. The third row illustrates the relative enhancement due to BC, calculated as $\Delta$ICNC/ICNC$_{noBC}$. In all maps, the TP region is outlined by a thick black boundary. **(b)** Seasonal cycle of mean ICNC (blue line) and the mean change due to BC INPs ($\Delta$ICNC, green line), spatially averaged over the TP and vertically averaged over altitudes where ice exists. The gray triangles show ICNC values from CloudSat averaged over 14 years reported by Chen et al. (2024). **(c)** Seasonal cycle of aerosol mass mixing ratios (mmr) averaged over the TP between 700 hPa and 100 hPa. The solid purple line shows the dust concentration, while the dashed pink line shows the BC concentration. Note that the BC concentration has been multiplied by a factor of 5 for improved visibility. **(d)** Seasonal cycle of column-integrated water path variables averaged over the TP. The plot shows the water vapor path (WVP, solid black line), ice water path (IWP, dotted purple line), and liquid water path (LWP, dotted orange line). Shaded regions in (b), (c) and (d) represent the standard deviation ($\pm 1\sigma$) for each variable.




**Figure 7. (a)** Spatial distribution of ICNC during the austral spring (SON) season over the South American outflow region (SAOR). The first row shows the ICNC from the simulation including BC as INPs at various pressure levels. The second row displays the absolute difference in ICNC (ΔICNC) between the simulations with and without BC as INPs. The third row illustrates the relative enhancement due to BC, calculated as $\Delta ICNC/ICNC_{noBC}$. In all maps, the SAOR region is outlined by a thick black boundary. **(b)** Seasonal cycle of mean ICNC (blue line) and the mean change due to BC INPs (ΔICNC, green line), spatially averaged over the SAOR and vertically averaged over altitudes where ice exists. **(c)** Seasonal cycle of aerosol mass mixing ratios (mmr) averaged over the SAOR between 700 hPa and 100 hPa. The solid purple line shows the dust concentration, while the dashed pink line shows the BC concentration. Note that the BC concentration has been multiplied by a factor of 5 for improved visibility. **(d)** Seasonal cycle of column-integrated water path variables averaged over SAOR. The plot shows the water vapor path (WVP, solid black line), ice water path (IWP, dotted purple line), and liquid water path (LWP, dotted orange line). Shaded regions in (b), (c) and (d) represent the standard deviation (±1σ) for each variable.





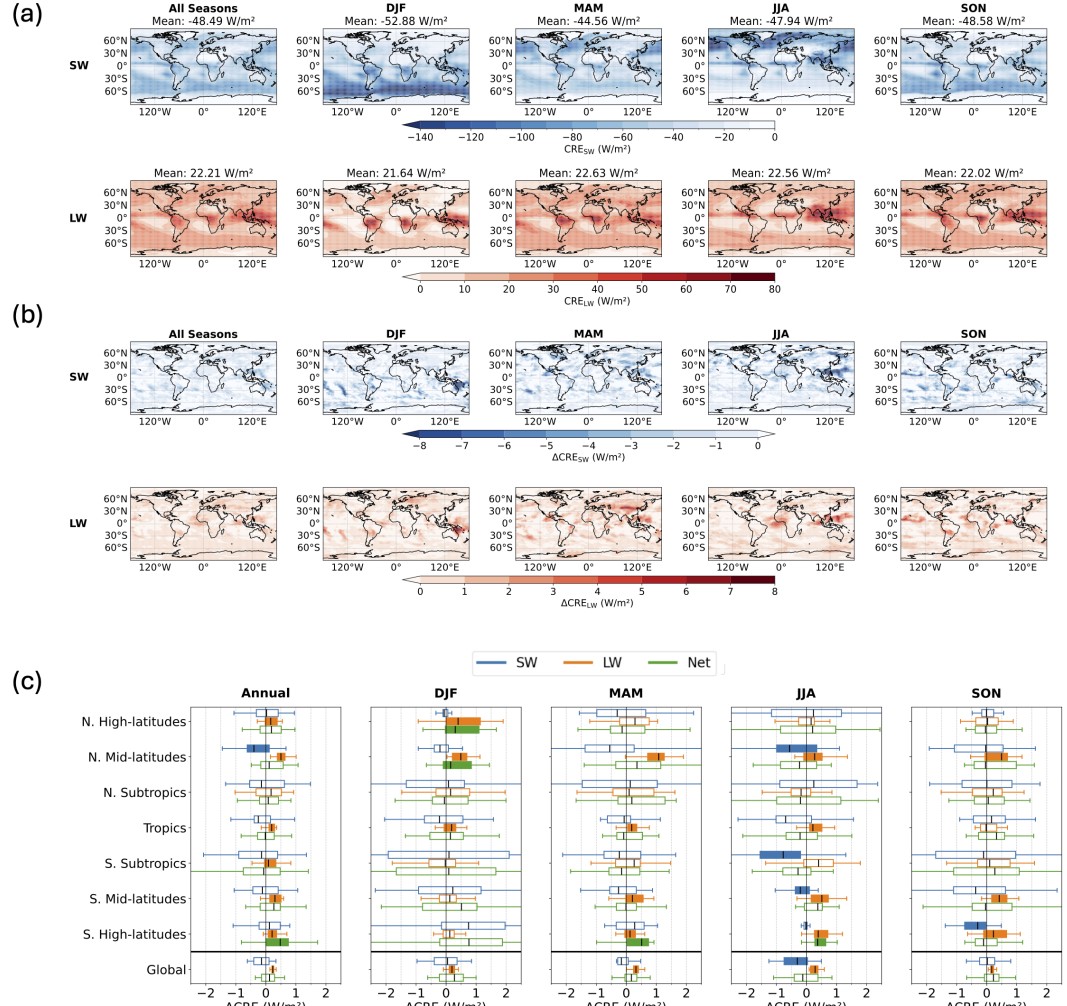

**Figure 8.** Cloud Radiative Effect (CRE) and its response to BC as INPs at the top of the atmosphere (TOA) computed over 2001–2020.
**(a)** Seasonal mean shortwave ($CRE_{SW}$, upper panel) and longwave ($CRE_{LW}$, lower panel) total cloud radiative effects from simulations that include BC INPs. Negative values (blue) indicate a net cooling effect by clouds, while positive values (red) indicate a net warming effect.
**(b)** The difference in CRE ($\Delta CRE$) caused by BC INPs, calculated as the difference between simulations with and without BC ($\Delta CRE = CRE_{BC} - CRE_{noBC}$). The upper and lower panels show the shortwave ($\Delta CRE_{SW}$) and longwave ($\Delta CRE_{LW}$) components, respectively. Here, negative values signify that BC enhances radiative cooling, while positive values signify an enhancement of radiative warming. **(c)** Box plots of annual and seasonal $\Delta CRE$ across different latitudinal regions, spatially averaged within each region and computed over a 20-year period. The regions shown are the N. High-latitudes (60°N–90°N), N. Mid-latitudes (35°N–60°N), N. Subtropics (23.5°N–35°N), Tropics (23.5°S–23.5°N), S. Subtropics (23.5°S–35°S), S. Mid-latitudes (35°S–60°S), S. High-latitudes (60°S–90°S), and the Global mean. Filled boxplots indicate that the mean $\Delta CRE$ is statistically significant different from 0 ($p < 0.05$).



*Code and data availability.* The AM4-MG2 source code used in this study can be found at https://github.com/NOAA-GFDL/AM4/tree/ MG2_xanadu_2020.02.01 and is also archived at https://doi.org/10.5281/zenodo.4313356. The data presented in this study are available upon request by email (contact: Xiaohan Li, `xiaohanl@princeton.edu`).

**Appendix A**

**Table A1.** Summary of parameters examined for the cirrus parcel model.

| Parameter | Unit | Values and binning |
| --- | --- | --- |
| Pressure (P) | hPa | 100, 200, 300, 400 |
| Temperature (T) | K | 190 to 233, linear, step = 3 K |
| Updraft velocity | cm s$^{-1}$ | 0.1 , then log-spaced from 1 to 30 (20 bins), log-spaced from 30 to 100 (20 bins) |
| Dust mass concentration | ng m$^{-3}$ | 1 , then log-spaced from 10 to 10,000 (15 bins) |
| Soot mass concentration | ng m$^{-3}$ | 1 , then log-spaced from 10 to 10,000 (15 bins) |
| Sulfate concentration | $\mu$g m$^{-3}$ | 0.01, 0.1, 1 |
| Sea salt mass concentration | $\mu$g m$^{-3}$ | 0.01, 0.1, 1 |





**Figure A1.** Total ice crystal number concentration ($N_{i,\text{tot}}$) and ice contributions from dust ($N_{i,\text{dust}}$) and black carbon (BC; $N_{i,\text{BC}}$) The figure maps ICNC as a function of cloud base temperature ($T$) and updraft velocity ($w$). The columns separate the total concentration from its dust and BC components, allowing for a direct comparison of their activity. The rows illustrate how these relationships change with decreasing altitude (increasing pressure from 100 to 400 hPa), highlighting the different temperature and updraft regimes where each nucleation pathway becomes dominant and how this dominance is modulated by the ambient pressure. All simulations shown were performed with fixed mass concentrations of 100 ng m$^{-3}$ for both dust and soot, and background concentrations of 0.1 $\mu$g m$^{-3}$ for both sea salt and sulfate.



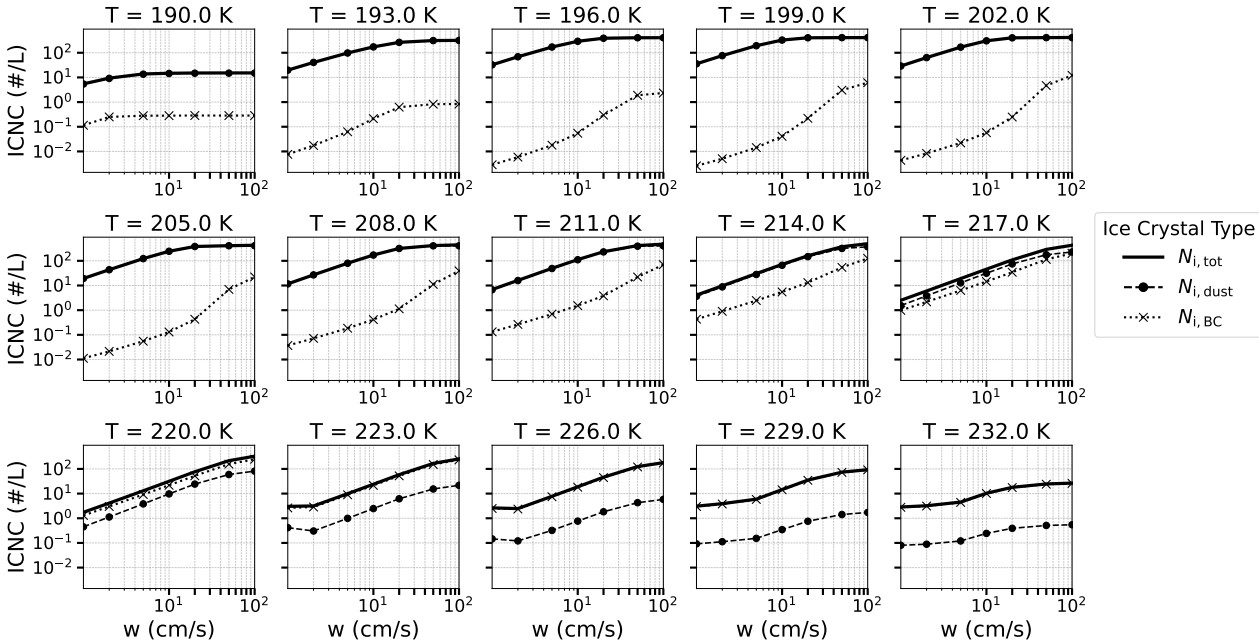

**Figure A2.** ICNC as a function of $w$. Each panel shows the ICNC dependence on $w$ for a specific cloud base temperature ($T$), as labeled in each subplot. The different line styles distinguish the total ICNC ($N_{i,\text{tot}}$, solid line), the contribution from dust nucleation ($N_{i,\text{dust}}$, circle-dashed line), and the contribution from BC nucleation ($N_{i,\text{BC}}$, cross-dotted line). All simulations were performed with fixed mass concentrations of 100 ng m$^{-3}$ for both dust and soot, and background concentrations of 0.1 $\mu$g m$^{-3}$ for both sea salt and sulfate at a cloud base pressure of 300 hPa.




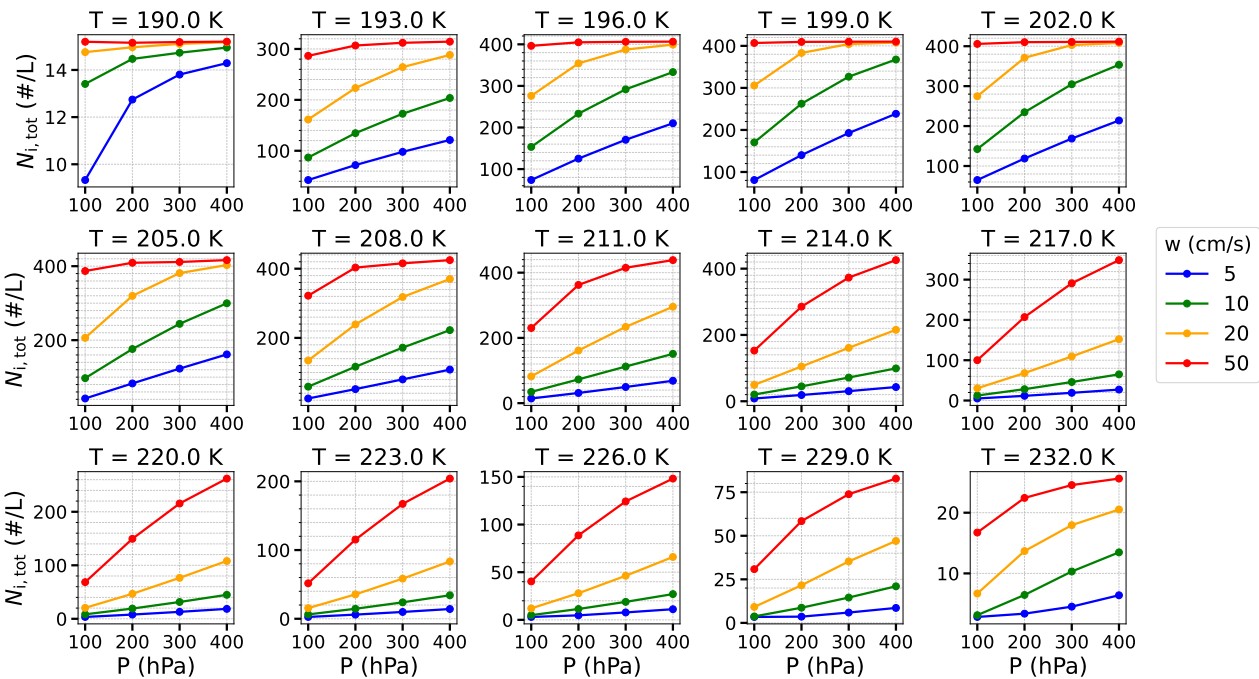

**Figure A3.** Total ice crystal number concentration ($N_{i,\mathrm{tot}}$) as a function of cloud base pressure ($P$). Each panel corresponds to a fixed cloud base temperature ($T$), as indicated in the subplot titles. Colored lines represent different updraft velocities ($w$). All simulations were performed with fixed mass concentrations of $100$ ng m$^{-3}$ for both dust and soot, and background concentrations of $0.1$ $\mu$g m$^{-3}$ for both sea salt and sulfate.

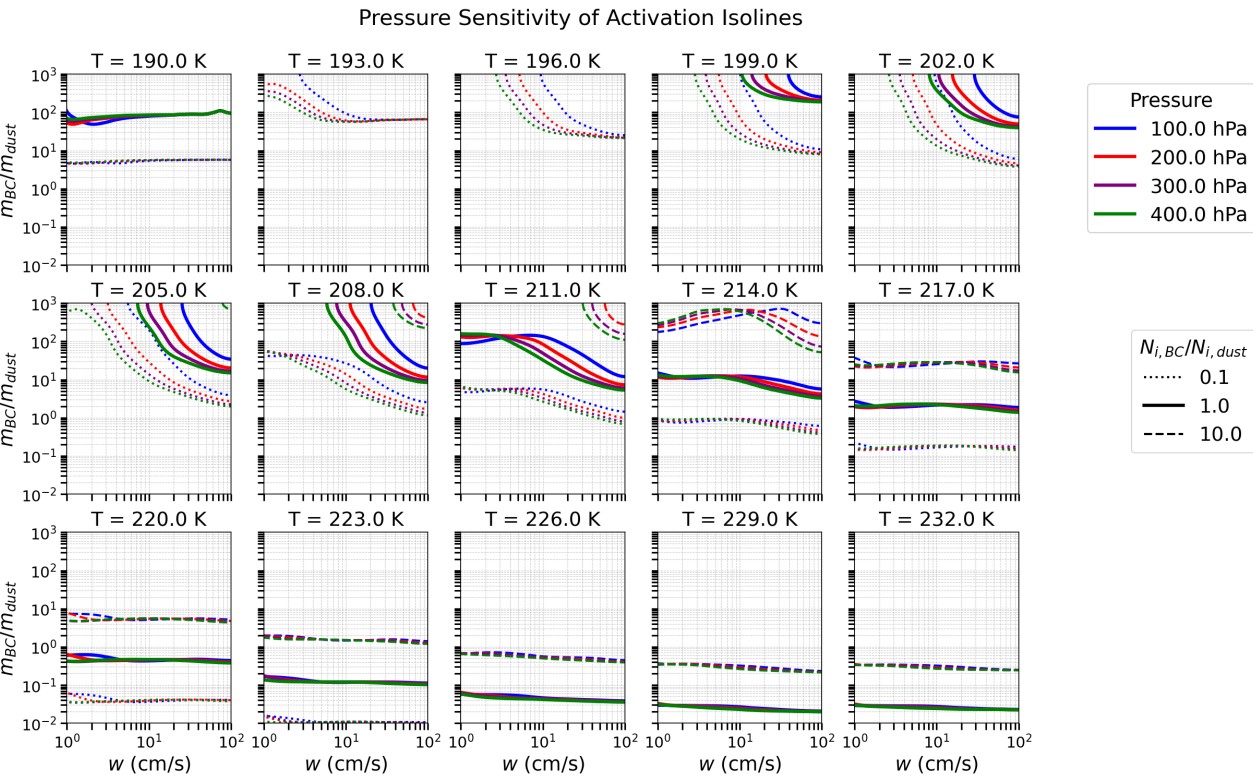

**Figure A4.** Sensitivity of the isolines of $N_{i,\mathrm{BC}}/N_{i,\mathrm{dust}}$ to meteorological conditions. Each subplot shows the ratio of BC-to-dust mass concentration ($m_{\mathrm{BC}}/m_{\mathrm{dust}}$) versus updraft velocity ($w$) at a different cloud base temperature $T$, as indicated in the subplot title. The lines represent isolines of $N_{i,\mathrm{BC}}/N_{i,\mathrm{dust}}$ equal to 0.1 (dotted), 1.0 (solid), and 10.0 (dashed). Different colors correspond to different cloud base pressures. All simulations assume fixed background concentrations of 0.1 $\mu$g m$^{-3}$ for both sea salt and sulfate.

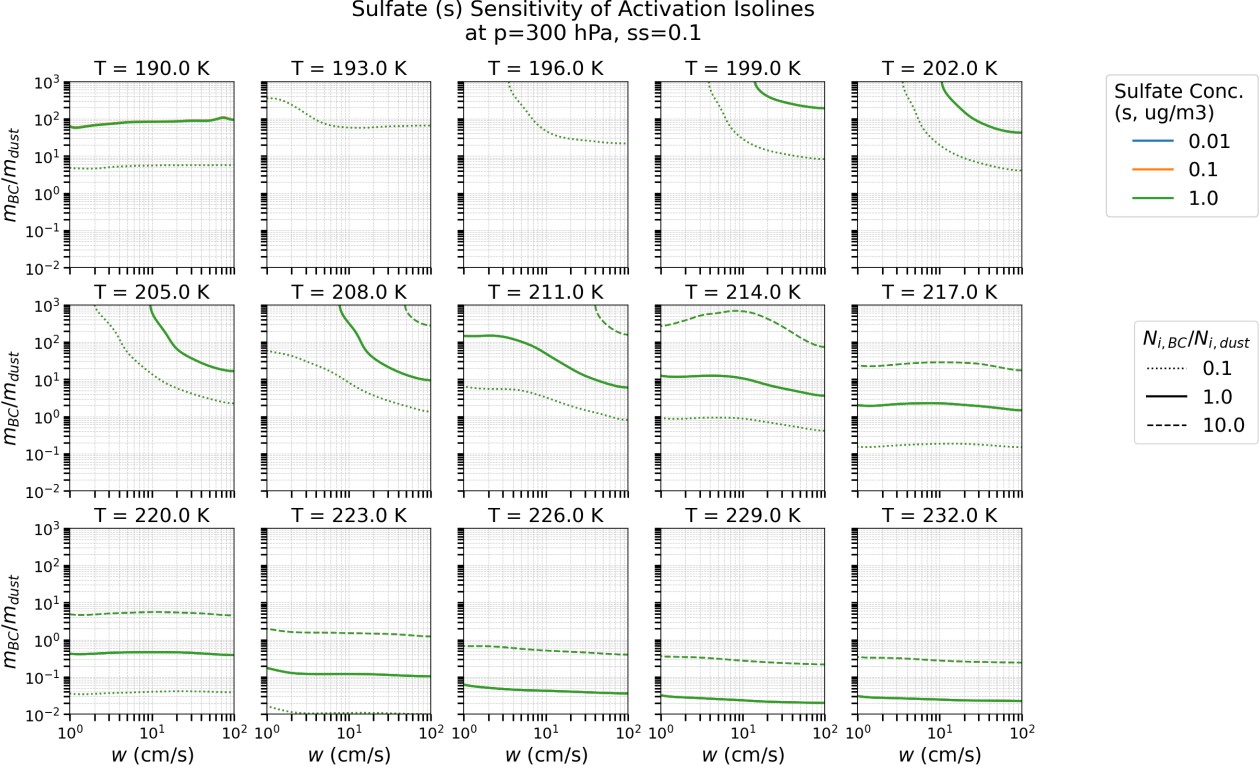

**Figure A5.** Sensitivity of the isolines of $N_{i,\mathrm{BC}}/N_{i,\mathrm{dust}}$ to sulfate concentration. Each subplot shows the ratio of BC-to-dust mass concentration ($m_{\mathrm{BC}}/m_{\mathrm{dust}}$) versus updraft velocity ($w$) at a different cloud base temperature $T$, as indicated in the subplot title. The lines represent isolines of $N_{i,\mathrm{BC}}/N_{i,\mathrm{dust}}$ equal to 0.1 (dotted), 1.0 (solid), and 10.0 (dashed). Different colors correspond to different sulfate concentrations. All simulations assume fixed background concentrations of 0.1 $\mu\mathrm{g}\ \mathrm{m}^{-3}$ for sea salt, dust, and soot. We note that the isolines are shown in green for all cases, as they overlap across different sulfate levels.



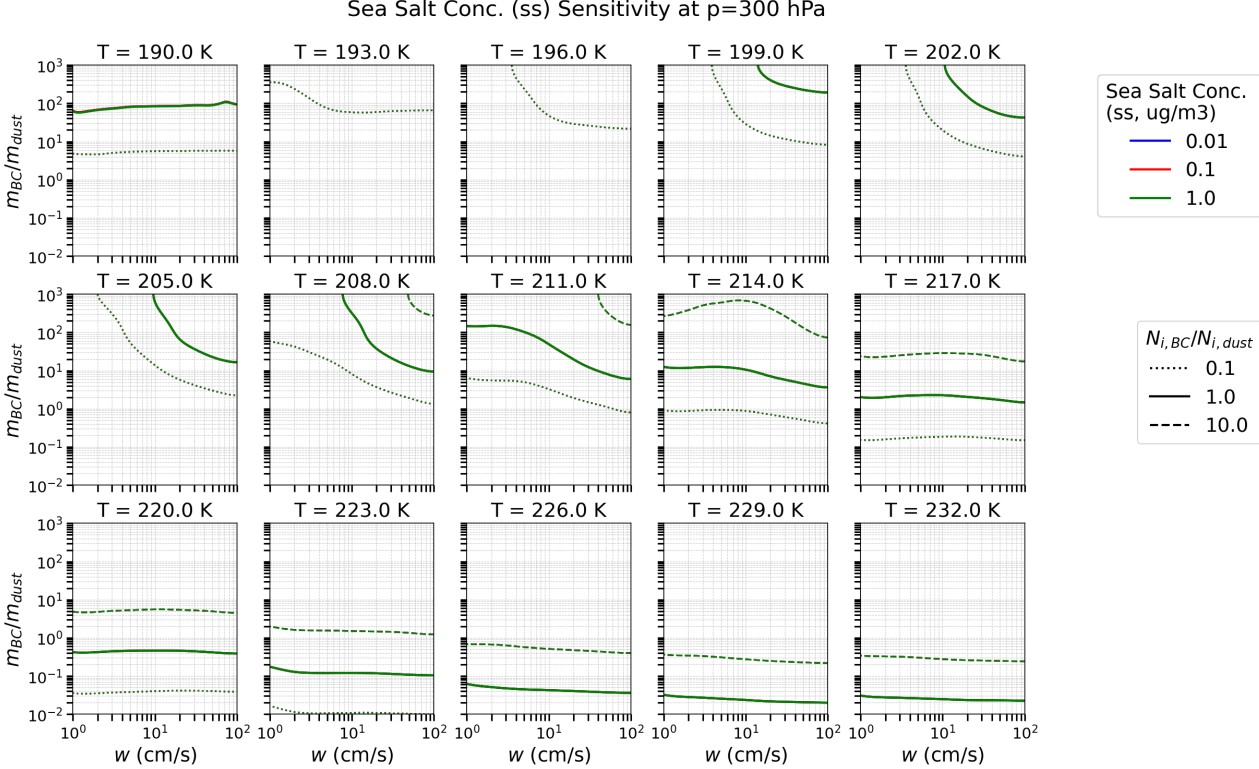

**Figure A6.** Sensitivity of the isolines of $N_{i,\mathrm{BC}}/N_{i,\mathrm{dust}}$ to seas salt concentration. Each subplot shows the ratio of BC-to-dust mass concentration ($m_{\mathrm{BC}}/m_{\mathrm{dust}}$) versus updraft velocity ($w$) at a different cloud base temperature $T$, as indicated in the subplot title. The lines represent isolines of $N_{i,\mathrm{BC}}/N_{i,\mathrm{dust}}$ equal to 0.1 (dotted), 1.0 (solid), and 10.0 (dashed). Different colors correspond to different sea salt concentrations. All simulations assume fixed background concentrations of 0.1 $\mu$g m$^{-3}$ for sulfate, dust, and soot. We note that the isolines are shown in green for all cases, as they overlap across different sea salt levels.



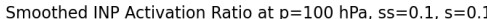

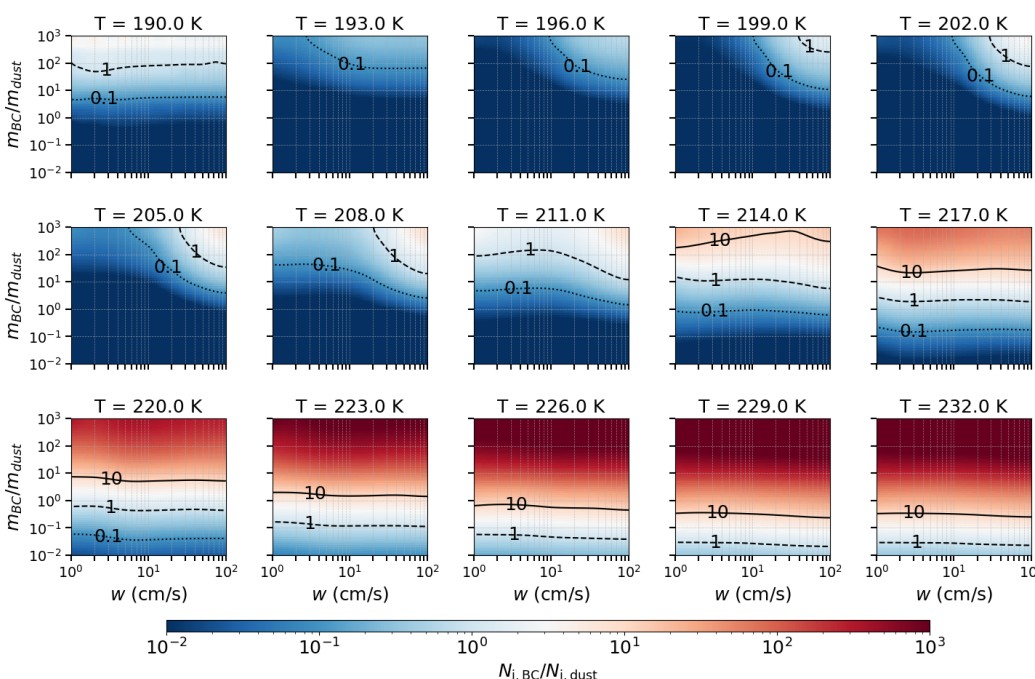

**Figure A7.** The relative importance of BC versus dust INPs, quantified by the ratio $N_{i,BC}/N_{i,dust}$ (color scale). The ratio is shown as a function of updraft velocity ($w$) and the initial aerosol mass ratio ($m_{BC}/m_{dust}$). Each panel corresponds to a different initial temperature from 190 K to 232 K. Isolines mark where the nucleation ratio is 0.1, 1, and 10. All simulations used a fixed pressure (100 hPa) and background aerosol (0.1 $\mu$g m$^{-3}$ sea salt and sulfate).

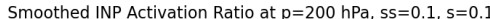

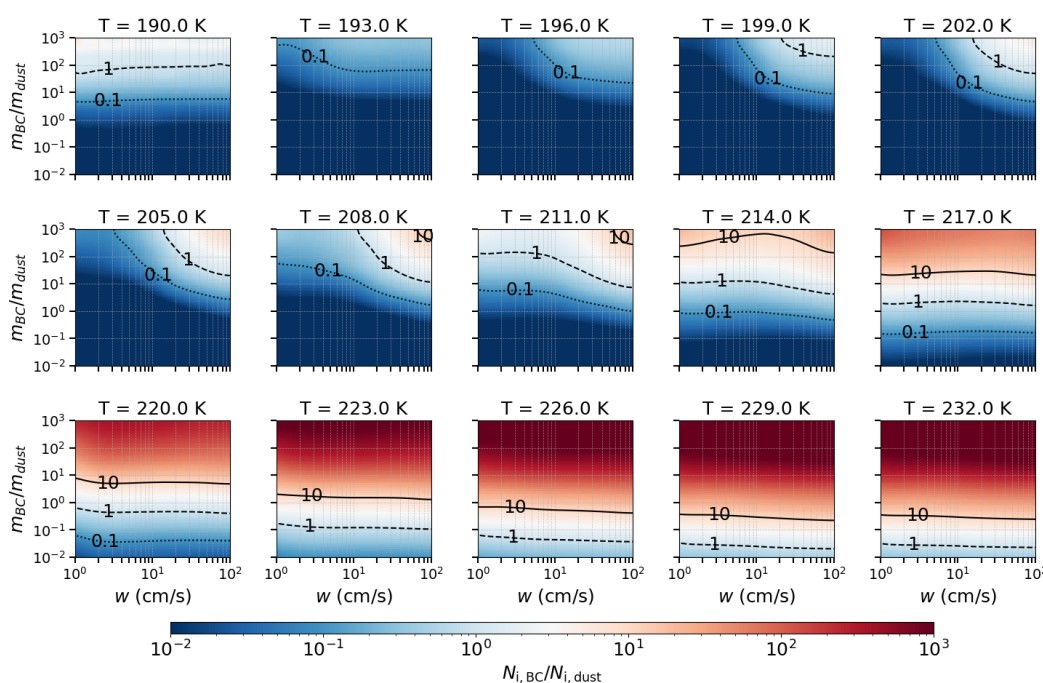

**Figure A8.** The relative importance of BC versus dust INPs, quantified by the ratio $N_{i,\mathrm{BC}}/N_{i,\mathrm{dust}}$ (color scale). The ratio is shown as a function of updraft velocity ($w$) and the initial aerosol mass ratio ($m_{\mathrm{BC}}/m_{\mathrm{dust}}$). Each panel corresponds to a different initial temperature from 190 K to 232 K. Isolines mark where the nucleation ratio is 0.1, 1, and 10. All simulations used a fixed pressure (200 hPa) and background aerosol (0.1 $\mu$g m$^{-3}$ sea salt and sulfate).



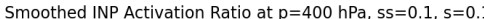

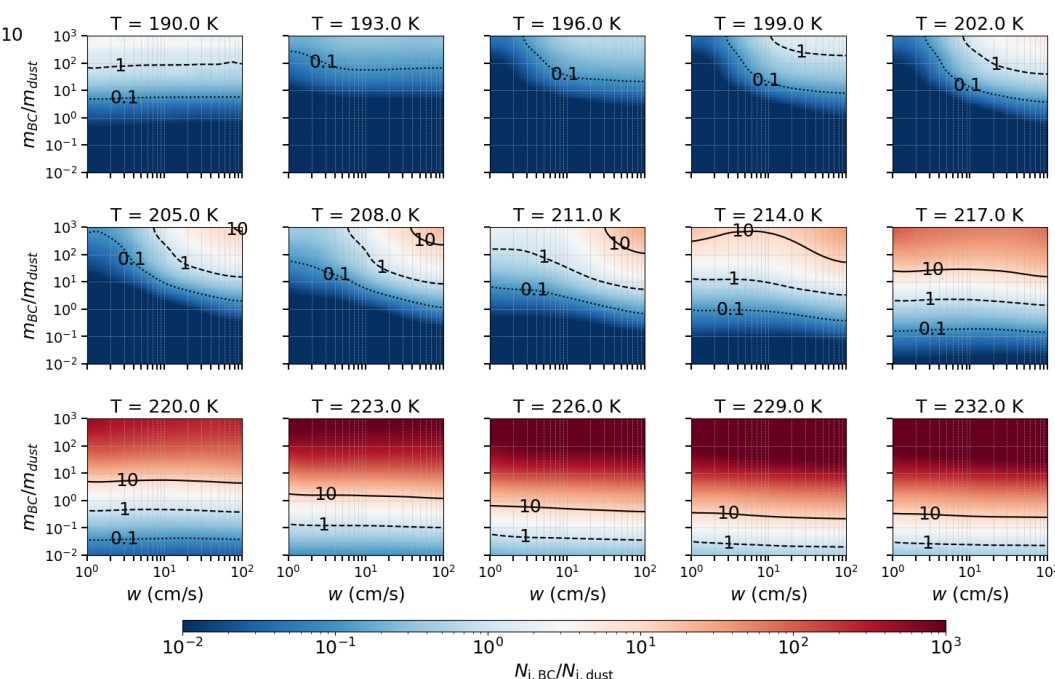

**Figure A9.** The relative importance of BC versus dust INPs, quantified by the ratio $N_{i,\mathrm{BC}}/N_{i,\mathrm{dust}}$ (color scale). The ratio is shown as a function of updraft velocity ($w$) and the initial aerosol mass ratio ($m_{\mathrm{BC}}/m_{\mathrm{dust}}$). Each panel corresponds to a different initial temperature from 190 K to 232 K. Isolines mark where the nucleation ratio is 0.1, 1, and 10. All simulations used a fixed pressure (400 hPa) and background aerosol (0.1 $\mu$g m$^{-3}$ sea salt and sulfate).





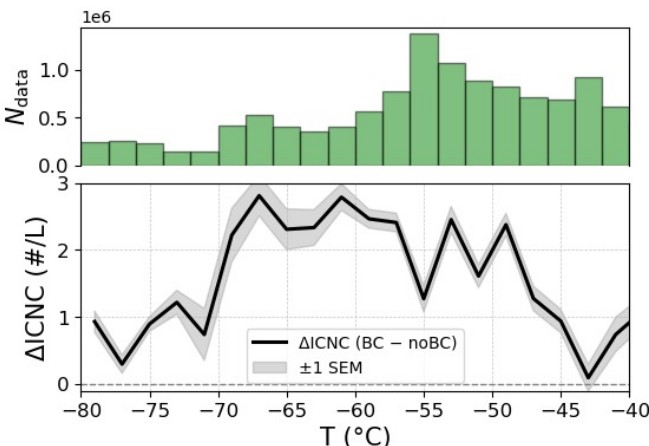

**Figure A10.** Difference in ICNC ($\Delta$ICNC) between GCM simulations with and without BC acting as INPs. The shaded region represents the standard error of the mean (SEM) of the difference. The top panel above shows the data count distribution versus temperature from the simulation that includes BC; the distribution for the no-BC case is omitted as it is nearly identical with the BC case.

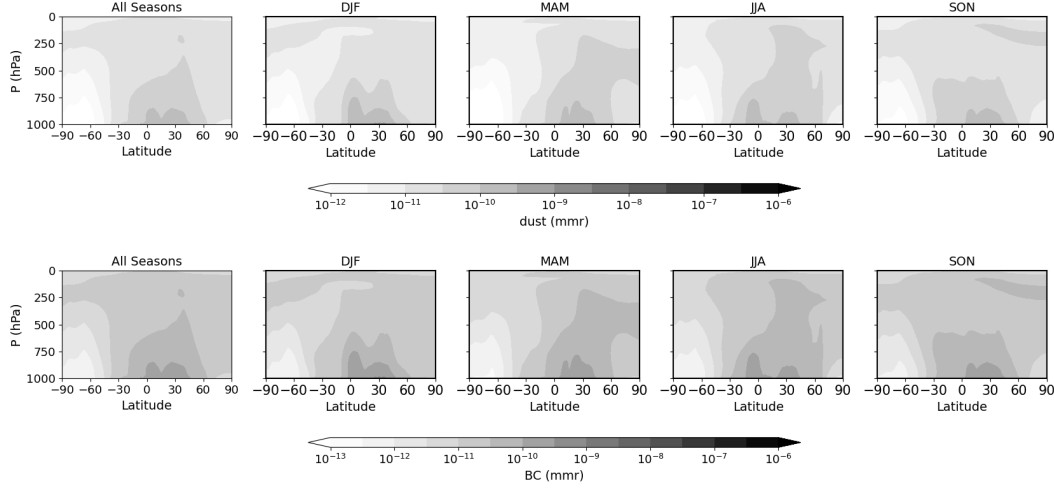

**Figure A11.** Seasonal zonal mean distributions of dust and black carbon (BC) concentration. The upper row of panels shows the mass mixing ratio (mmr) for dust, while the lower row shows the distribution for black carbon. Each column represents a different season.





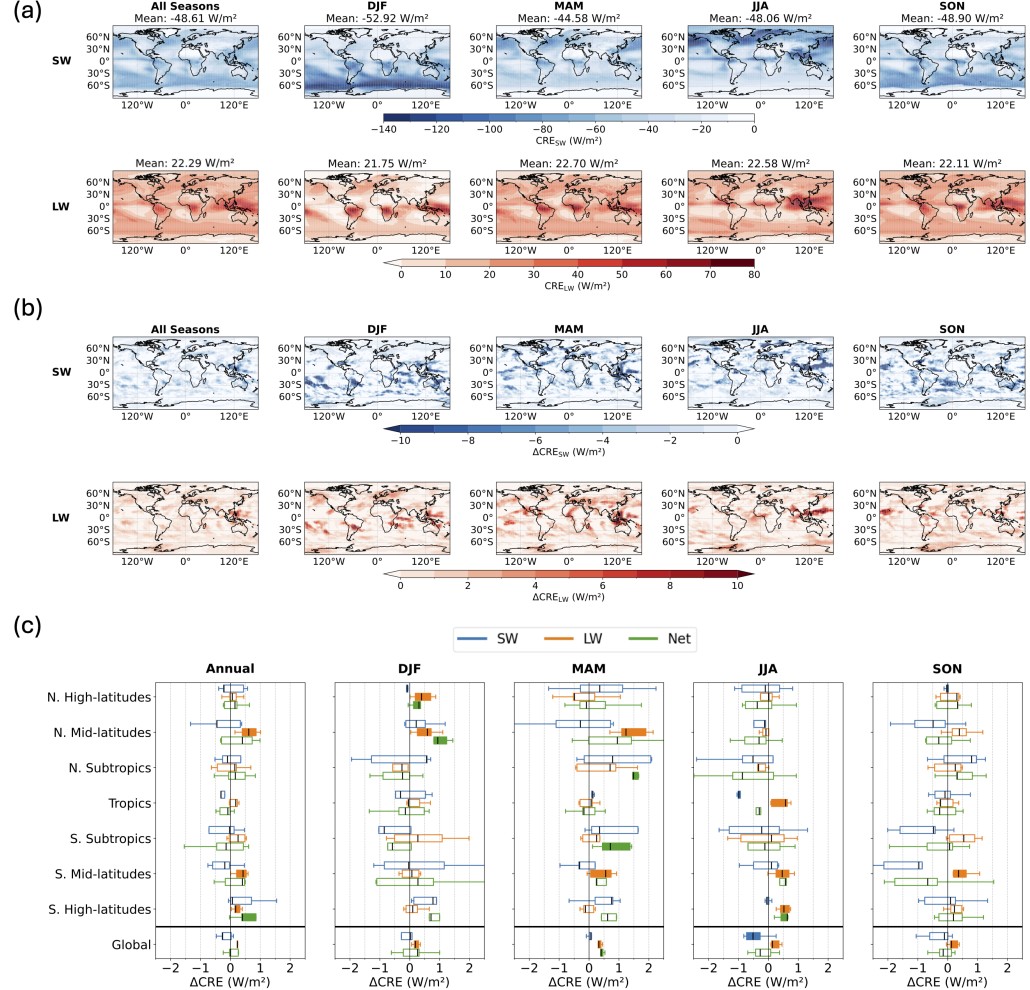

**Figure A12.** Cloud Radiative Effect (CRE) and its response to BC as INPs at the top of the atmosphere (TOA) computed over 2001–2005. **(a)** Seasonal mean shortwave ($CRE_{SW}$, upper panel) and longwave ($CRE_{LW}$, lower panel) cloud radiative effects from simulations that include BC INPs. Negative values (blue) indicate a net cooling effect by clouds, while positive values (red) indicate a net warming effect. **(b)** The difference in CRE ($\Delta CRE$) caused by BC INPs, calculated as the difference between simulations with and without BC ($\Delta CRE = CRE_{BC}$ - $CRE_{noBC}$). The upper and lower panels show the shortwave ($\Delta CRE_{SW}$) and longwave ($\Delta CRE_{LW}$) components, respectively. Here, negative values signify that BC enhances radiative cooling, while positive values signify an enhancement of radiative warming. **(c)** Box plots of annual and seasonal $\Delta CRE$ across different latitudinal regions, spatially averaged within each region and computed over a 5-year period. The regions shown are the N. High-latitudes (60°N–90°N), N. Mid-latitudes (35°N–60°N), N. Subtropics (23.5°N–35°N), Tropics (23.5°S–23.5°N), S. Subtropics (23.5°S–35°S), S. Mid-latitudes (35°S–60°S), S. High-latitudes (60°S–90°S), and the Global mean. Filled boxplots indicate that the mean $\Delta CRE$ is statistically significant different from 0 ($p < 0.05$).



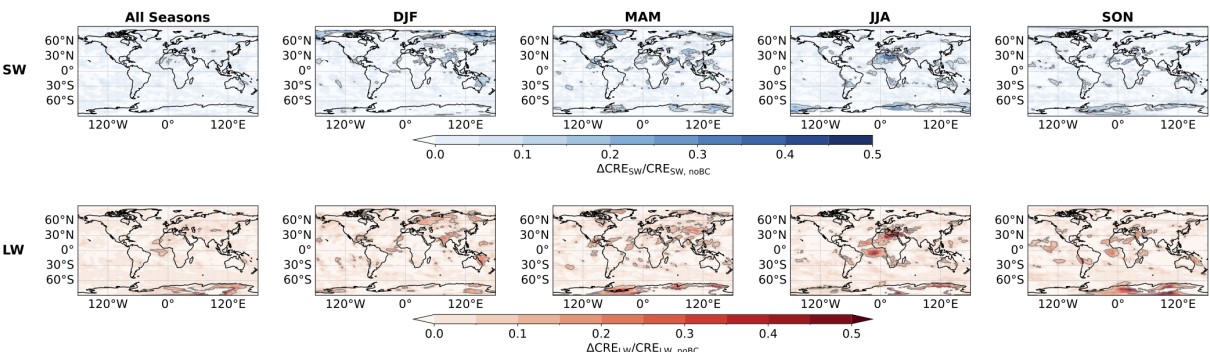

**Figure A13.** The relative change in cloud radiative effect (CRE) due to BC INPs, expressed as $\Delta CRE/CRE_{noBC}$ averaged over 2001-2020. The upper and lower panels show the shortwave ($\Delta CRE_{SW}/CRE_{noBC,SW}$) and longwave ($\Delta CRE_{LW}/CRE_{noBC,LW}$) components, respectively. Isolines indicate enhancement fractions equal to 0.1.

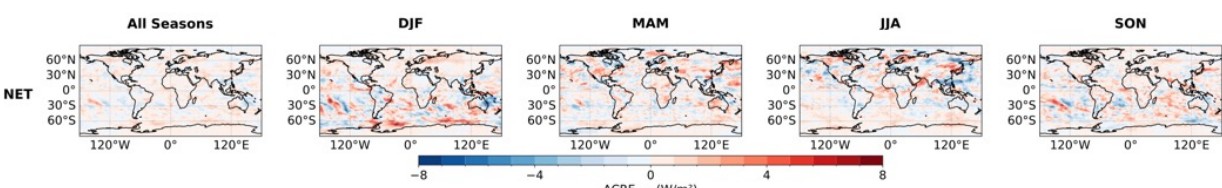

**Figure A14.** The spatial distribution of the net cloud radiative effect ($\Delta CRE_{net}$) induced by BC INPs averaged over 2001-2020. Results are shown for the annual mean and for different seasons. Positive values indicate that the longwave (LW) warming effect dominates the shortwave (SW) cooling effect.



*Author contributions.* X.L.: Conceptualization, Methodology, Investigation, Formal Analysis, Software, Data Curation, Visualization, Writing. S.F.: Supervision, Conceptualization, Methodology, Software, Data Curation, Review & Editing. H.G.: Conceptualization, Methodology, Software, Review & Editing. P.G.: Supervision, Conceptualization, Review & Editing.

*Competing interests.* The authors declare no competing financial interest.

*Disclaimer.* The contents of this article are solely the responsibility of the authors and do not represent the official views of any agency or institution.

*Acknowledgements.* This research was supported by the National Oceanic and Atmospheric Administration, U.S. Department of Commerce under Award NA23OAR4320198. The statements, findings, conclusions, and recommendations are those of the authors and do not necessarily reflect the views of the National Oceanic and Atmospheric Administration, or the U.S. Department of Commerce. The authors would like to
thank Fabien Paulot, Larry Horowitz, John Dunne, Ming Zhao, and Jing Feng for insightful suggestions and discussions on this manuscript.



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
