# Peer review of "Resolving the roles of soot and dust in cirrus cloud ice formation at regional and global scales: insights from parcel and climate models"

_EGUsphere, 2025_

## Author Comment (AC1)

**ATMOSPHERIC AND OCEANIC SCIENCES**

DEPARTMENT OF GEOSCIENCES

PRINCETON UNIVERSITY, PRINCETON, NEW JERSEY 08544

Nov 19th, 2025

Dear ACP Editor and Reviewer,

Thank you for your feedback and careful reading of our manuscript. We are happy to submit a revised manuscript, "*Resolving the roles of soot and dust in cirrus cloud ice formation at regional and global scales: insights from parcel and climate models*". We highly appreciate your time and valuable suggestions. Below, we provide detailed responses to all reviewer comments. For clarity, the reviewer comments are presented in gray, our responses in **black**, and revised text in **blue**. We hope that our revisions address all concerns.

Best wishes,
Xiaohan Li, Songmiao Fan, Huan Guo, Paul Ginoux

**Summary of Content:**
Page 2-9: Reply to Reviewer 1

**Reply to Reviewer 1**

Reviewer: 1
This paper examines the roles of soot (black carbon, BC) and mineral dust as ice-nucleating particles (INPs) in shaping cirrus cloud properties and their global radiative impacts. The authors first use a cloud parcel model to perform 5.5 million simulations under a wide range of conditions, including variations in cloud-base temperature, pressure, updraft velocity, and aerosol (dust, soot, sulfate, and sea salt) mass concentrations. These simulations provide process-level insights into how soot and dust influence ice formation in cirrus clouds. The results are then incorporated into a global climate model to assess large-scale effects. The study finds that soot increases global mean ice crystal number concentration (ICNC) by about 5%, with regional enhancements of up to 90% in the upper troposphere. Furthermore, BC INPs strengthen the global longwave cloud radiative effect and lead to a statistically significant net warming during polar winters in both hemispheres. Overall, I appreciate the comprehensive simulations and analyses conducted by the authors. The results might provide valuable insights into the influence of BC INPs on cirrus cloud formation and radiative forcing at both global and regional scales. However, I think major revisions are needed to especially clarify the model configuration so that readers can better understand and assess the simulation results.

**Reply to reviewer #1 summary.** Thanks for your careful reading of our manuscript. We deeply appreciate your valuable comments and suggestions. Below you will find our replies to your comments.

Major comments:
Section 2.1. Some key model configurations are either missing or insufficiently described. I recommend that the authors explicitly present these details, so readers can clearly understand the model framework and better interpret the results later presented in this paper.

Line 103. What are the corresponding number concentrations? Number concentration has a more direct connection to ice number concentration.

**Reply to Q1.1:** Thank you for raising this point. We agree that number concentration provides a more direct connection to ice number concentration. The corresponding number concentrations are as follows: for soot, a mass concentration of 1–10,000 ng m$^{-3}$ corresponds to $4.0 \times 10^5$–$4.0 \times 10^9$ m$^{-3}$; for dust, $4.2 \times 10^3$–$4.2 \times 10^7$ m$^{-3}$; for sulfate, 10–1,000 ng m$^{-3}$ corresponds to $1.3 \times 10^6$–$1.3 \times 10^8$ m$^{-3}$; and for sea salt, $3.5 \times 10^4$–$3.5 \times 10^6$ m$^{-3}$. We have revised the text in the first paragraph of Section 2.1.1 to include this information. The updated text now reads:

"To examine aerosol-cloud interactions, four distinct aerosol types were simulated: soot, dust, sulfate, and sea salt. A total of 15 mass concentrations of soot and dust were specified, corresponding to number concentrations of $4 \times 10^5$ to $4 \times 10^9$ m$^{-3}$ for soot and $4.2 \times 10^3$ to $4.2 \times 10^7$ m$^{-3}$ for dust. For sulfate and sea salt, three mass concentrations were specified, ranging from 10 to 1,000 ng m$^{-3}$, corresponding to number concentrations of $1.3 \times 10^6$ to $1.3 \times 10^8$ m$^{-3}$ for sulfate and $3.5 \times 10^4$ to $3.5 \times 10^6$ m$^{-3}$ for sea salt."

Line 105. What are the ranges of initial RH_ice and RH_w? I'm curious to know whether cirrus cloud starts to form at the beginning of the simulation or form after the parcel reaches a certain altitude.

**Reply to Q1.2:** Thank you for raising this question. The initial relative humidity over ice (RH_ice in the parcel model is set to 1.1. For temperatures below –40 °C, this corresponds to relative humidity with respect to water (RH_w) values below approximately 0.75. One would need to define a critical ice number concentration to say "cirrus cloud starts to form". In our parcel model, the ice-nucleated number fractions for dust and soot are calculated as a function of ice supersaturation and temperature following Ullrich et al. (2017), with INAS scaling factors from Ullrich et al. (2019).

We have revised the text of in the first paragraph of Section 2.1.1 to include the initial condition information:

"The parcel model computes pressure ($P$) and temperature ($T$) as an air parcel ascends from its initial state under adiabatic conditions, with the initial relative humidity over ice ($RH_{ice}$) set to 1.1. For temperatures below –40 °C, this corresponds to relative humidity with respect to water ($RH_w$) values below approximately 0.75. Additionally, the model calculates $RH_{ice}$, $RH_w$ and the number concentrations and sizes of droplets and ice crystals during the parcel expansion."

And added the following discussion on cirrus formation at the end of Section 2.1.1:

"We note that a critical ice number concentration must be defined to determine when a cirrus cloud starts to form. In our parcel model, rather than explicitly tracking cirrus formation, we calculate the ice-nucleated number fractions for dust and soot as a function of ice supersaturation and temperature as discussed in Section 2.1.2."

Since there are sulfate and sea salt, can supercooled liquid droplets formed in the parcel model if the air is supersaturated with respect to water?

**Reply to Q1.3:** Yes, as noted in the second paragraph of section 2.1.1 we do consider the supercooled liquid droplets formed in the parcel model, such as "the deliquescent sulfate and sea salt aerosols, as well as liquid droplets formed when the diffusion of water molecules to deliquescent aerosols leads to rapid growth, reaching the critical supersaturation over water".

Lines 108-123. What are the parameterization equations used in the parcel model for homogeneous and heterogeneous nucleation (soot, dust, sulfate, sea salt)? Please add them in the main text or supplementary materials.

Lines 124-133. Please add the ice growth equation used in the parcel model in the main text or SI.

Section 2.1.2. List parameterization equations for INAS used in the parcel model.

**Reply to Q1.4/Q1.5/Q1.6:** Thank you for these helpful suggestions. In our parcel model, for heterogeneous ice nucleation, the deposition nucleation is parameterized following Ullrich et al. (2017, 2019), and the immersion freezing for dust is parameterized following Alpert and Knopf (2016). For homogeneous nucleation, we follow the parameterization from Koop et al. (2000). The ice crystal growth equation follows Pruppacher and Klett (1998), and the INAS formulation is taken from Ullrich et al. (2017). We have now explicitly included these parameterization equations by adding a new section in Appendix A as "**Section A1 Parameterization of nucleation and ice growth processes in the parcel model**" for clarity and reproducibility. A reference to this section has also been added at the end of Section 2.1.2 as:

"The detailed formulations of the INAS density, homogeneous and heterogeneous nucleation rates, and ice crystal growth can be found in Appendix A1**."**

And below is the content in Section A1:

**Appendix A**

**A1    Parameterization of nucleation and ice growth processes in the parcel model**

515    In our parcel model, homogeneous ice nucleation is parameterized following Koop et al. (2000). Heterogeneous ice nucleation is represented by deposition freezing parameterizations from Ullrich et al. (2017, 2019) and immersion freezing parameterizations for dust from Alpert and Knopf (2016). Ice crystal growth follows Pruppacher and Klett (1998), and the ice-nucleating active surface site (INAS) formulation is adopted from Ullrich et al. (2017). The explicit expressions for these parameterizations are summarized below.

520    **A1.1 Homogeneous nucleation.** We follow the water-activity-based parameterization of Koop et al. (2000). The homogeneous nucleation rate $J_{\text{hom}}$ (in cm$^{-3}$s$^{-1}$) is calculated as a function of the difference in water activity, $\Delta a_w$.

The nucleation rate $J_{\text{hom}}$ is given by the polynomial fit:

$$\log_{10}(J_{\text{hom}}) = -906.7 + 8502(\Delta a_w) - 26924(\Delta a_w)^2 + 29180(\Delta a_w)^3 \tag{A1}$$

Where $\Delta a_w$ is the difference between the water activity of the aqueous solution ($a_w$, which is assumed to be in equilibrium

525    with the environment, $a_w = \text{RH}_w/100$) and the water activity at the ice melting point ($a_{i,w}$):

$$\Delta a_w = a_w - a_{i,w} \tag{A2}$$

**A1.2 INAS density parameterization for heterogeneous deposition freezing.** We use the ice nucleation active site (INAS) density parameterization framework from Ullrich et al. (2017). As shown in Equation 1, the number of ice crystals nucleated, $N_i$, is calculated based on the available aerosol surface area, $S_{\text{aer}}$, and the INAS density, $n_s(T, S_i)$, which is a function of

530    temperature ($T$) and ice saturation ratio ($S_i$). The $n_s$ parameterizations differ for each nucleation mode and aerosol type:

**(a) Deposition nucleation (mineral dust).** For deposition nucleation on mineral dust, $n_s$ (in m$^{-2}$) is described by the "U-shaped" function of $T$ (in K) and $S_i$:

$$n_s(T, S_i) = \exp\left\{ \alpha(S_i - 1)^{1/4} \cos[\beta(T - \gamma)]^2 \frac{\text{arccot}[\kappa(T - \lambda)]}{\pi} \right\} \tag{A3}$$

The fit parameters for mineral dust (from Ullrich et al., 2017) are: $\alpha = 285.692$, $\beta = 0.017$, $\gamma = 256.692$ K, $\kappa = 0.080$ K$^{-1}$,

535    and $\lambda = 200.745$ K. This parameterization is valid for temperatures between 206 K and 240 K.

**(b) Deposition nucleation (soot).** For deposition nucleation on soot, the same functional form (Equation A3) is used. For soot with an organic carbon content of less than or equal 20 wt%, the fit parameters are: $\alpha = 46.021$, $\beta = 0.011$, $\gamma = 248.560$ K, $\kappa = 0.148$ K$^{-1}$, and $\lambda = 237.570$ K. This parameterization is valid for temperatures between 195 K and 235 K. However, the

**24**

fit parameters $(\alpha, \beta, \gamma, \kappa, \lambda)$ are also depend on the soot's organic carbon content, leading to a shift toward higher $S_i$ for soot

540 with higher organic carbon content as detailed in (Ullrich et al., 2017).

**(c) Scaling for coated aerosols.** To account for the suppression of nucleation efficiency by coatings (e.g., sulfate or organics), we apply scaling factors based on Ullrich et al. (2019). The INAS density ($n_s$) from the parameterizations above is multiplied by a factor to represent this effect. Based on their findings, the $n_s$ for coated mineral dust is scaled by a factor of 0.05, and for coated soot by a factor of 0.01.

545 **A1.3 Immersion freezing (mineral dust).** For immersion freezing of mineral dust, we follow the stochastic, water activity-based immersion freezing model (ABIFM) framework developed by Alpert and Knopf (2016), which accounts for the time-dependent, stochastic nature of immersion freezing. This approach calculates the heterogeneous ice nucleation rate coefficient, $J_{het}(T, a_w)$ (in $cm^{-2}s^{-1}$), based on classical nucleation theory. Where the number of ice crystals ($N_i$) for an aerosol population with surface area $A_{aer}$ and aerosol number $N_{aer}$ is given by:

550
$$N_i = N_{aer}(1 - \exp(-J_{het} \cdot A_{aer}t)) \tag{A4}$$

The nucleation rate $J_{het}$ is parameterized as a function of temperature $T$ and water activity $a_w$ as

$$\log_{10}(J_{het}) = m[a_w(T) - a_{w, ice}(T)] + c \tag{A5}$$

and

$$a_{w, ice}(T) = p_{ice}(T)/p_w(T) \tag{A6}$$

555 where $p_{ice}(T)$ and $p_w(T)$ are the saturation water vapor pressure of ice and pure liquid water at temperature T. The parameters of $m = 22.62$ and $c = -1.35$ are used for natural dust.

**A1.4 Ice crystal growth.** The diffusional growth of an individual ice crystal (mass $m_i$) is calculated based on the standard equation for vapor diffusion and heat conduction, as found in **Pruppacher and Klett (1998)**:

$$\frac{dm_i}{dt} = \frac{4\pi C_i(S_i - 1)}{F_d + F_k} \tag{A7}$$

560 where $C_i$ is the capacitance of the ice crystal, which accounts for its non-spherical shape; $S_i$ is the ice saturation ratio of the environment; $F_d$ and $F_k$ are the "thermodynamic" (heat conduction) and "diffusion" (vapor diffusion) resistance terms, and

$$F_d = \frac{L_s}{K_a T}\left(\frac{L_s}{R_v T} - 1\right) \tag{A8}$$

$$F_k = \frac{R_v T}{D_v p_{s,i}} \tag{A9}$$

565 where $L_s$ is the latent heat of sublimation; $K_a$ is the thermal conductivity of air; $T$ is the ambient temperature; $R_v$ is the gas constant for water vapor; $D_v$ is the diffusivity of water vapor in air; $p_{s,i}$ is the saturation vapor pressure over ice.

**25**

Line 149. People might not be familiar with "the U-shaped curves". Need to add more explanations or rephrase this sentence.

**Reply to Q1.7:** Thanks for pointing this. For clarification, we have revised our expression to avoid the use of "the U-shaped curves" as follows:

"The deposition nucleation ns isolines for desert dust show a minimum in the ice saturation ratio–temperature (Si–T) diagram at an intermediate temperature below 240 K. At temperatures below this minimum, the required Si increases as temperature decreases (a negative slope), which can be explained by classical nucleation theory. Conversely, at temperatures above this minimum, the required Si also increases as temperature increases (a positive slope), a behavior likely caused by a pore condensation and freezing mechanism. The deposition nucleation measured for soot at temperatures below 240 K exhibits a similar pattern to that of desert dust, but with isolines shifted toward higher Si for soot with higher organic carbon content."

It is not clear to me what microphysical scheme is used in the parcel model, Lagrangian, bin, or bulk?

**Reply to Q1.8:** Thank you for raising this question. In our parcel model, dry aerosols are distributed into prescribed size bins, while the activated droplets and nucleated ice crystals are treated in a Lagrangian framework—that is, each particle is individually tracked following its activation or nucleation and subsequent diffusional growth. This hybrid approach allows us to explicitly resolve particle size distributions while accurately representing the microphysical evolution of droplets and ice crystals. We have added the following description at the end of the first paragraph in section 2.1.1 as follows:

"We note that in the parcel model, dry aerosols are distributed into prescribed size bins, while activated droplets and ice crystals are tracked individually in a Lagrangian framework. This approach explicitly resolves particle size distributions and captures the detailed microphysical evolution of droplets and ice crystals."

Section 2.2. Need more details about how parcel model results are implemented in AM4-MG2.

Line 188. "Within the GCM at each time step". What is the time step, 30 minutes (physical time step) or 2.5 minutes (dynamic core)?

**Reply to Q2.1:** Thank you for pointing this out. The time step we refer to is the *physical timestep* of 30 min. We have clarified this in the revised text as follows:

"Within the GCM at each time step (i.e., the physical timestep of 30 min), this lookup table is queried to determine Ni,dust and Ni,BC when the ambient temperature is below 233.15 K."

Line 192. "the GCM interpolates linearly for pressure and temperature, and logarithmically for updraft velocity and the aerosol mass concentration". Are there any conditions that you need to extrapolate values outside the ranges of the box model?

**Reply to Q2.2:** Thanks for raising this point. Yes, it is possible for the modeled pressure, temperature, updraft velocity, or aerosol mass concentration to fall outside the bounds of the lookup table. In such cases, the model constrains these variables to the nearest boundary of the table rather than performing extrapolation. This approach is justified because INP concentrations are physically negligible at the lower boundaries and approach saturation (or maximum parameterized values) at the upper boundaries. This method also ensures numerical stability by avoiding potential artifacts from extrapolation. We have added the following clarification in the revised manuscript under Section 2.2.2 at the end of the paragraph:

"We note that, in certain cases, the model-simulated pressure, temperature, updraft velocity, or aerosol mass concentration may exceed the range represented in the lookup table. In such cases, the model constrains these variables to the nearest upper or lower limit of the table rather than performing extrapolation. This approach is

justified because INP concentrations are physically negligible near the lower boundaries and approach saturation near the upper boundaries. This treatment also ensures numerical stability by avoiding potential artifacts from extrapolation."

More details are needed to explain how parcel model results are implemented in AM4-MG2. My understanding is that if you know the updraft velocity, pressure, temperature, and mass concentrations of dust, soot, sulfate, and sea salt, you can calculate the number concentration of ice crystals nucleated on dust and black carbon from a lookup take based on the parcel model results. Are those ice crystals diagnostic or prognostic variables? Will the formed ice particles have the feedback on temperature and water vapor in AM4-MG2? The lookup take is based on simulation results at 2.5 minutes or 30 minutes? If cirrus cloud already exists in What about the ice crystal size? It is from the lookup table or assigned in AM4-MG2?

**Reply to Q2.3:** Thanks for raising this question. Both ice crystal number and mass concentrations are prognostic. The formed ice particles will release latent heat and deplete water vapor in AM4-MG2. As addressed in the previous question, the lookup take is based on simulation results at 30 min. The ice crystal size is determined by the ice crystal number and mass concentrations under the assumption of Gamma distribution in AM4-MG2. In the revised manuscript. We have provided more details as below in the second paragraph under Section 2.2.1:
"AM4-MG2 explicitly prognoses both the mass mixing ratios and number concentrations for four hydrometeor types: cloud water, cloud ice, rain, and snow. The treatment of ice nucleation is critical for modeling mixed-phase clouds, as it serves as the primary source of ice crystal number concentration. For mixed-phase clouds, a temperature- and dust-dependent ice nucleation scheme is applied (Fan et al., 2019), while for cirrus clouds, the nucleated ice number concentration is derived from parcel model simulations, as described in Section 2.1. Assuming that ice crystals follow Gamma size distributions, their mean size is determined from the ice crystal number and mass concentrations. The nucleation of ice crystals is coupled with the depletion of water vapor and the release of latent heat, both of which are represented in the MG2 scheme (Morrison and Gettelman, 2008; Gettelman and Morrison, 2015a). Furthermore, to ensure consistency between the prognostic treatments of ice crystal number and mass concentrations, AM4-MG2 includes the detrainment of ice number concentration from convection to large-scale clouds, following the approach of Kristjansson et al. (2000). The model also considers the shortwave and longwave radiative effects of precipitating hydrometeors (rain and snow)."

Figure 1. It seems that homogeneous ice nucleation is ignorable, am I correct? Is it because water vapor is consumed by the formation and growth of ice particles formed by heterogeneous ice nucleation on soot and dust?

**Reply to Q3:** Thank you for raising this question. Homogeneous ice nucleation is not entirely negligible in our parcel model. We do observe homogeneous nucleation under warmer conditions (typically for temperatures above ~230 K) and when dust and BC concentrations are low.

To illustrate the role of homogeneous nucleation, we computed the fraction of homogeneous ice crystals, defined as

$$f_{\text{homo}} = \frac{N_{i,\text{tot}} - N_{i,\text{dust}} - N_{i,\text{BC}}}{N_{i,\text{tot}}}$$

and plotted it as a function of updraft velocity at 232 K and 300 hPa, using sea salt and sulfate concentrations of 100 ng m-3 for various combinations of dust and BC mass concentrations (new Figure A10). Figure A10 shows that when dust and BC concentrations are low, homogeneous nucleation contributes up to ~96% of total ice crystals. As expected, this fraction decreases as INP concentrations increase (from left to right panels).

[Figure]

Figure A3: Fraction of ice crystals formed by homogeneous nucleation f_homo as a function of updraft velocity w at 232 K and 300 hPa, with sulfate and sea-salt concentrations fixed at 100 ng/m-3. Panels show different combinations of dust and BC mass concentrations in unit of ng/m$^{-3}$ (from low to high, left to right). When dust and BC concentrations are low, homogeneous nucleation contributes more than 90% of total ice crystals. The contribution decreases with increasing INP concentrations at the same updraft velocity, consistent with the expected suppression of homogeneous nucleation by heterogeneous nucleation on dust and soot.

To illustrate the temperature dependence, we also plotted the maximum value of $f_{\mathrm{homo}}$ across all updraft velocities as a function of temperature (new Figure A11). Figure A11 demonstrates that homogeneous nucleation becomes important only at warmer temperatures (typically $T \gtrsim 225$ K). At colder temperatures, the maximum homogeneous fraction is very small (<1%), consistent with the dominance of heterogeneous nucleation on dust and soot at these conditions. These additions clarify that homogeneous nucleation does occur in our parcel model, but its contribution becomes negligible at colder cirrus temperatures due to the rapid consumption of supersaturation by heterogeneous ice formation and growth.

[Figure]

Figure A4: Maximum fraction of homogeneous ice crystals $f_{\mathrm{homo}}$ across all simulated updraft velocities as a function of temperature with a initial pressure of 300 hPa and sulfate and sea-salt mass concentrations of 100 ng/m-3.

Homogeneous nucleation becomes important only at warmer temperatures (T >~230 K), where the maximum value of $f_{\text{homo}}$ can be substantial. At colder temperatures, the homogeneous fraction is negligible (<1%), reflecting the strong depletion of supersaturation by heterogeneous nucleation on dust and soot. The different color lines show different combinations of dust and BC mass concentrations in unit of ng/m$^{-3}$

To clarify this homogeneous nucleation behavior, we have added the above two Figures to the manuscript, and added the discussion in the first paragraph of Section 3.1.2 as:

"We note that homogeneous nucleation is not completely negligible in our parcel model. Our results show that under warmer cirrus conditions (typically for temperatures above 230 K) and when dust and BC concentrations are low, the fraction of ice crystals formed by homogeneous nucleation, defined as f_homo = (N_i,tot-N_i,dust-N_i,BC)/N_i,tot, can reach values as high as ~96%. The dependence of f_homo on INP concentration, temperature, and updraft velocity is shown in Figures A3 and A4."

Figure 5. What might be the reason for the negative value of Delta_ICNC in b?

**Reply to Q4:** Thank you for this question. Several processes may contribute to the negative values of ΔICNC in panel (b). One likely explanation is an indirect dynamical–microphysical effect: enhanced ice formation at higher altitudes in the simulation with BC INPs can deplete water vapor that would otherwise be transported downward, thereby suppressing local ice nucleation at lower altitudes and producing a negative ΔICNC. In addition, negative values can arise under conditions where homogeneous nucleation dominates—specifically when soot is neglected and dust concentrations are sufficiently low for homogeneous nucleation to occur. In such cases, adding BC INPs can shift the balance between heterogeneous and homogeneous nucleation, leading to a net reduction in ICNC at certain temperatures.

Is semi-direct effect of BC considered in AM4-MG2?

**Reply to Q5:** Yes, AM4-MG2 includes the semi-direct effect of all absorbing aerosols, mostly BC, but also dust and to a minor extent, OC. In AM4-MG2, this semi-direct effect arises from aerosol absorption of solar radiation, which heats the atmosphere (both within and outside clouds) and can lead to cloud evaporation. The model supports both all-sky and clear-sky radiative calculations, and in all cases the semi-direct effect is represented through absorption-induced atmospheric heating. We have added the following text in the second paragraph of Section 2.2.1 to demonstrate this point:

"In addition, AM4-MG2 includes the semi-direct effect of all absorbing aerosols, with BC as the primary contributor and additional contributions from dust and, to a lesser extent, organic aerosols. In AM4-MG2, this semi-direct effect arises from aerosol absorption of solar radiation, which heats the atmosphere both within and outside clouds and can promote cloud evaporation. The model supports both all-sky and clear-sky radiative calculations, and in all cases the semi-direct effect is represented through absorption-induced atmospheric heating."

Minor comments:

Figure 1. Since BC is used to represent soot in this study, change "C_m,soot" to "C_m,BC" in the figure, the caption, and the text (e.g., line 215).

*Line 144: should it be "ice nucleation active surface site density"?*

K is used in Figure 2 and degree C is used in Figure 3. Please be consistent with the units used in the figure and main text.

*Table A1: "sulfate concentration"->"sulfate mass concentration"?*

Figure A11. The unit of BC is "mmr". Change it to "ng m-3"?

**Reply to Q6:** Thanks! All the text and Figures have been revised accordingly.

---

## Author Comment (AC2)

**ATMOSPHERIC AND OCEANIC SCIENCES**

DEPARTMENT OF GEOSCIENCES

PRINCETON UNIVERSITY, PRINCETON, NEW JERSEY 08544

Nov 19th, 2025

Dear ACP Reviewer,

Thank you for your feedback and careful reading of our manuscript. We are happy to submit a revised manuscript, "*Resolving the roles of soot and dust in cirrus cloud ice formation at regional and global scales: insights from parcel and climate models*". We highly appreciate your time and valuable suggestions. Below, we provide detailed responses to all reviewer comments. For clarity, the reviewer comments are presented in gray, our responses in **black**, and revised text in **blue**. We hope that our revisions address all concerns.

Best wishes,
Xiaohan Li, Songmiao Fan, Huan Guo, Paul Ginoux

**Summary of Content:**
Page 2-5: Reply to Reviewer 2

**Reply to Reviewer 2**

Reviewer: 2
This study uses a parcel model to find relationships on the competition between homogeneous and heterogeneous nucleation based upon a large variety of black carbon and dust concentration. The resulting findings are inserted into the GFDL AM4-MG2 model to examine how black carbon and mineral dust impacts cirrus clouds globally. Black carbon enhances global ice crystal number concentration and significantly increases the annual global longwave cloud radiative effect, causing warming, particularly during polar winters. The results emphasize the roles of soot and dust and the need to assess the climate impact of increasing wildfire emissions on atmospheric ice processes. The paper is very well written, and conclusions are well supported. I have only minor comments.
**Reply to reviewer #2 summary.** Thanks for your careful reading of our manuscript. We deeply appreciate your valuable comments and suggestions. Below you will find our replies to your comments.

In the abstract, line 8: I like to reserve the word observe to real observations. I wonder if the sentence could be rephrased to something like "The strongest enhancements are found during boreal"
Page 5, line 147: Add "of" ....where N_aer is the number density of ice nucleating....
Page 7, line 209. ICNC has only been defined in the abstract, but not in the main text yet.
**Reply to Q1:** Thanks for raising the above comments. We have revised the text as suggested and added the definition of ICNC (ice crystal number concentration) in the last paragraph of the Introduction to ensure clarity for readers.

Page 4, lines 118-123: The diameters and standard deviations of the aerosol size distribution are fixed. Could varying the size and shape of the size distributions change your results?
**Reply to Q2:** We thank the reviewer for this insightful question. Indeed, varying the size and shape of the aerosol size distribution could potentially influence the results. However, as described in the methodology, AM4-MG2 employs a bulk aerosol scheme in which only the aerosol mass concentration is prognosed with the mean particle size and geometric standard deviation are fixed and prescribed. This assumption of fixed aerosol size and distribution is a common practice in GCM simulations, including those participating in CMIP5 and CMIP6. We acknowledge that aerosol size can influence activation and ice nucleation processes, but given the large uncertainties in observed size distributions and to maintain consistency with GCM representations, our parcel model experiments also adopt prescribed, fixed size parameters. The effects of varying aerosol size can be investigated once a fully coupled aerosol microphysics scheme becomes available, which is currently under active development at GFDL. We have added the following text in the second paragraph of Section 2.1.1 to acknowledge the impact of size distribution and limitations:
"We note that a fixed aerosol size distribution is used in this study, although varying the size and shape of the distribution could potentially influence the results. The choice of fixed size parameters is primarily to maintain consistency with the bulk aerosol scheme in the host climate model AM4-MG2, where only the aerosol mass concentration is prognosed, and the mean particle size and geometric standard deviation are prescribed. This bulk representation is a common practice in GCM simulations, including those participating in CMIP5 and CMIP6. We acknowledge that aerosol size can influence activation and ice nucleation processes; however, given the large uncertainties in observed size distributions and to ensure consistency with GCM representations, our parcel model experiments also adopt prescribed, fixed size parameters. The effects of varying aerosol size will be explored in future studies once a fully coupled aerosol microphysics scheme becomes available, which is currently under active development."

Page 7, line 199. It is stated that the AMIP simulation is run through 2006. Why is the analysis period only to 2005 and not through 2006?
**Reply to Q3:** Thank you for pointing this out. The AMIP simulation was initialized in 2000 and extended into early 2006 only to generate model restart files. Therefore, the complete simulation years available for analysis are from 2000 to 2005. To avoid confusion, we have revised the text as follows:
"The simulation was initialized in 2000 and run to the end of 2006, with the first year treated as model spin-up, and the following 5 years of 2001–2005 for analysis."

Section 3.1.2 You describe the ICNC dependence on aerosol composition. But there is no mention about the competition between homogeneous and heterogeneous dependence on aerosol composition. I am not sure how to present it, but it would be interesting to have a figure showing how the change in composition impacts homogeneous freezing.

**Reply to Q4:** Homogeneous ice nucleation is not entirely negligible in our parcel model. We do observe homogeneous nucleation under warmer cirrus conditions (typically for temperatures above ~230 K) and when dust and BC concentrations are sufficiently low.

To illustrate the role of homogeneous nucleation, we computed the fraction of homogeneous ice crystals, defined as

$$f_{\text{homo}} = \frac{N_{i,\text{tot}} - N_{i,\text{dust}} - N_{i,\text{BC}}}{N_{i,\text{tot}}}$$

and plotted it as a function of updraft velocity at 232 K and 300 hPa, using sea salt and sulfate concentrations of 100 ng m-3 for various combinations of dust and BC mass concentrations (new Figure A10). Figure A10 shows that when dust and BC concentrations are low, homogeneous nucleation contributes up to ~96% of total ice crystals. As expected, this fraction decreases as INP concentrations increase (from left to right panels).

[Figure]

Figure A3: Fraction of ice crystals formed by homogeneous nucleation f_homo as a function of updraft velocity w at 232 K and 300 hPa, with sulfate and sea-salt concentrations fixed at 100 ng/m-3. Panels show different combinations of dust and BC mass concentrations in unit of ng/m$^{-3}$ (from low to high, left to right). When dust and BC concentrations are low, homogeneous nucleation contributes up to more than 90% of total ice crystals. The contribution decreases with increasing INP concentrations at the same updraft velocity, consistent with the expected suppression of homogeneous nucleation by heterogeneous nucleation on dust and soot.

To illustrate the temperature dependence, we also plotted the maximum value of $f_{\text{homo}}$ across all updraft velocities as a function of temperature (new Figure A11). Figure A11 demonstrates that homogeneous nucleation becomes important only at warmer temperatures (typically $T \gtrsim 225$ K). At colder temperatures, the maximum homogeneous fraction is very small (<1%), consistent with the dominance of heterogeneous nucleation on dust and soot at these conditions. These additions clarify that homogeneous nucleation does occur in our parcel model, but its contribution becomes negligible at colder cirrus temperatures due to the rapid consumption of supersaturation by heterogeneous ice formation and growth.

[Figure]

Figure A4: Maximum fraction of homogeneous ice crystals $f_{homo}$ across all simulated updraft velocities as a function of temperature. Homogeneous nucleation becomes important only at warmer temperatures (typically $T \gtrsim 225$ K), where the maximum value of $f_{homo}$ can be substantial. At colder temperatures, the homogeneous fraction is negligible (<1%), reflecting the strong depletion of supersaturation by heterogeneous nucleation on dust and soot. The different color lines show different combinations of dust and BC mass concentrations in unit of ng/m$^{-3}$

To clarify this competition between homogeneous and heterogeneous nucleation, we have added the above two Figures to the manuscript, and added the discussion in the first paragraph of Section 3.1.2 as:

"We note that homogeneous nucleation is not completely negligible in our parcel model. Our results show that under warmer cirrus conditions (typically for temperatures above 230 K) and when dust and BC concentrations are low, the fraction of ice crystals formed by homogeneous nucleation, defined as f_homo = (N_i,tot-N_i,dust-N_i,BC)/N_i,tot, can reach values as high as ~96%. The dependence of f_homo on INP concentration, temperature, and updraft velocity is shown in Figures A3 and A4."

Page 11, Figure 3. The In-situ Heymsfield symbols are difficult to see, special in the green region. The dotted lines could also be slightly enhanced. It would also be interesting to see how the model performs without the BC in this figure.
**Reply to Q5:** Thank you for this comment. We have reduced the transparency of the in-situ Heymsfield observations so that the symbols are more visible, particularly in regions with dense model output. We have also slightly enhanced the dotted lines for improved clarity.

Regarding the suggestion to include the model results without BC directly in Figure 3, we appreciate the idea; however, the background density map differs substantially between the BC and no-BC simulations because the underlying ICNC distributions shift in magnitude and structure. Overlaying the two density fields in the same panel would therefore not allow a meaningful or visually interpretable comparison. Instead, as shown in the Appendix, we provide a figure displaying the difference in ICNC between the BC and no-BC simulations as a function of temperature. This allows a clear comparison of the BC contribution while avoiding overlapping density biases.

Conclusion: Lines 480-485. The authors have based their conclusion on one dust and BC parameterization. There are other parameterizations out there that can lead to different results. For example, how is the temperature dependent active site density in other parameterizations and how could this impact the results?
**Reply to Q6:** We thank the reviewer for highlighting this important point regarding the uncertainties associated with different ice nucleation parameterizations. We agree that using alternative formulations could influence the results and

we plan to collaborate with others to compare different parameterizations and assess their impacts on simulated cirrus properties. We also note that the parameterization from Ulrich et al, 2017, used in this study, is based on multiple dust and soot samples measured in the AIDA chamber, thus providing a representative, though not exhaustive dataset. We have added the following sentences in the last paragraph of the conclusion part to acknowledge the uncertainties arising from different parameterizations:

"It should also be noted that other parameterizations for the ice-nucleating ability of dust and soot exist beyond those applied in this study, and alternative formulations may yield different results. Future collaborative efforts to intercompare parameterizations and quantify their impacts on simulated cirrus properties would therefore be valuable. "

Figures 5 and 8 are very small and hard to read on printed paper.
**Reply to Q7:** Thank you for pointing this out. We have updated Figures 5 and 8 by increasing their size and adjusting the layout to improve readability, especially when printed.

---

## Author Response (AR2)

**ATMOSPHERIC AND OCEANIC SCIENCES**

DEPARTMENT OF GEOSCIENCES

PRINCETON UNIVERSITY, PRINCETON, NEW JERSEY 08544

Dec 7th, 2025

Dear ACP Editor,

Thank you for your feedback and careful reading of our manuscript. We are happy to submit a revised manuscript, "*Resolving the roles of soot and dust in cirrus cloud ice formation at regional and global scales: insights from parcel and climate models*". We highly appreciate your time and valuable suggestions. Below, we provide detailed responses to all reviewer comments. For clarity, the reviewer comments are presented in gray, our responses in **black**, and revised text in **blue**. We hope that our revisions address all concerns.

Best wishes,
Xiaohan Li, Songmiao Fan, Huan Guo, Paul Ginoux

**Reply to the Editor**

Thank you for submitting a revision of the manuscript entitled "Resolving the roles of soot and dust in cirrus cloud ice formation at regional and global scales: insights from parcel and climate models" to Atmospheric Chemistry and Physics. I have two received two reviews of your revised manuscript. Both reviewers recommend publication of the manuscript in the present form. Based on the recommendation of the reviewers and my own review of the manuscript, I recommend acceptance of your article in Atmospheric Chemistry and Physics pending the following few minor revisions:

1. Please revise the title and line 82 of the manuscript to "a climate model" (singular, not plural) as GFDL AM4-MG2 is the only climate model used in the study.

3. Lines 121-123: please reference the heterogeneous nucleation schemes for dust and soot used in the parcel model.

**Reply to Q1&Q3:** Thanks for raising these points. We have revised the manuscript accordingly as suggested.

2. Please include a description of the results pertaining to the role of dust in cirrus cloud formation in the Abstract. The Abstract currently focuses on soot.

**Reply to Q2:** We thank the Editor for this helpful suggestion. We have we have revised the abstract to incorporate a description of the dust-related results, ensuring that the roles of both dust and soot in cirrus formation are represented. The current abstract reads:

"Atmospheric aerosols can serve as ice-nucleating particles (INPs), influencing cirrus cloud formation and properties. While mineral dust is recognized as an effective INP, the role of soot remains less explored, limiting climate impact assessments. Here we use cloud parcel model simulations to examine the competitive ice nucleation behavior of soot and dust, alongside homogeneous nucleation. **These process-level simulations reveal that dust dominates heterogeneous ice nucleation at colder temperatures (T < 210 K),** whereas soot becomes effective at warmer temperatures (T > 215 K), particularly when dust concentrations are low or under strong updrafts.  To evaluate their global-scale implications, we integrate these results into the GFDL AM4-MG2 climate model. We find that **dust shapes the baseline spatial and seasonal ice crystal number concentration (ICNC) patterns,** while soot (represented in the model as black carbon, BC) enhances global-mean ICNC by ~5%. However, BC-driven increases in ICNC can be much larger in the upper troposphere (500–250 hPa), reaching up to 90%. The strongest enhancements are found during boreal spring across Eurasia and the Maritime Continent, and during austral spring over South America and the South Atlantic. Radiatively, BC INPs can enhance the annual global longwave cloud radiative effect by approximately 0.24 W m$^{-2}$ and cause statistically significant net warming in both polar regions during their respective winters. These results highlight the coupled roles of dust and soot in cloud ice formation, underscoring the need to assess the impacts of rising wildfire emissions on atmospheric ice processes and associated climate effects."

---

## Author Response (AR3)

Dec 8th, 2025

Dear Ivy,

Thank you for your careful review and helpful suggestion regarding the title. We understand the concern that the original plural form ('models') might inadvertently imply the use of multiple global climate models. To fully address this ambiguity while maintaining grammatical balance, we have replaced 'models' with 'modeling.' The revised title is: "**Resolving the roles of soot and dust in cirrus cloud ice formation at regional and global scales: insights from parcel and climate modeling**".

We hope this revised wording more accurately reflects the methodology and resolves the ambiguity.

Best wishes,
Xiaohan Li, Songmiao Fan, Huan Guo, Paul Ginoux